# Subtype and cell type specific expression of lncRNAs provide insight into breast cancer

Sunniva Stordal Bjørklund[1,2], Miriam Ragle Aure [1,2], Jari Häkkinen [3], Johan Vallon-Christersson[3], Surendra Kumar [2,4], Katrine Bull Evensen[2], Thomas Fleischer [2], Jörg Tost [5], OSBREAC*, Kristine K. Sahlberg[2,6], Anthony Mathelier [1,7], Gyan Bhanot [8,9], Shridar Ganesan[9], Xavier Tekpli [1,2,25✉] & Vessela N. Kristensen [1,10,25✉]

Long non-coding RNAs (lncRNAs) are involved in breast cancer pathogenesis through chromatin remodeling, transcriptional and post-transcriptional gene regulation. We report robust associations between lncRNA expression and breast cancer clinicopathological features in two population-based cohorts: SCAN-B and TCGA. Using co-expression analysis of lncRNAs with protein coding genes, we discovered three distinct clusters of lncRNAs. In silico cell type deconvolution coupled with single-cell RNA-seq analyses revealed that these three clusters were driven by cell type specific expression of lncRNAs. In one cluster lncRNAs were expressed by cancer cells and were mostly associated with the estrogen signaling pathways. In the two other clusters, lncRNAs were expressed either by immune cells or fibroblasts of the tumor microenvironment. To further investigate the cis-regulatory regions driving lncRNA expression in breast cancer, we identified subtype-specific transcription factor (TF) occupancy at lncRNA promoters. We also integrated lncRNA expression with DNA methylation data to identify long-range regulatory regions for lncRNA which were validated using ChiA-Pet-Pol2 loops. lncRNAs play an important role in shaping the gene regulatory landscape in breast cancer. We provide a detailed subtype and cell type-specific expression of lncRNA, which improves the understanding of underlying transcriptional regulation in breast cancer.

[1] Department of Medical Genetics, Oslo University Hospital, Oslo, Norway. [2] Department of Cancer Genetics, Institute for Cancer Research, Oslo University Hospital Radiumhospitalet, Oslo, Norway. [3] Division of Oncology, Department of Clinical Sciences Lund, Lund University, Medicon Village, SE 22381 Lund, Sweden. [4] Department of Ocean Sciences, Memorial University of Newfoundland, St. John's, NL, Canada. [5] Laboratory for Epigenetics and Environment, Centre National de Recherche en Génomique Humaine, CEA – Institut de Biologie Francois Jacob, University Paris Saclay, Evry, France. [6] Department of Tumor Biology, Institute for Cancer Research, Oslo University Hospital, 4950 Oslo, Norway. [7] Centre for Molecular Medicine Norway (NCMM), Nordic EMBL Partnership, University of Oslo, 0349 Oslo, Norway. [8] Department of Physics and Astronomy, Rutgers University, Piscataway, NJ 08854, USA. [9] Rutgers Cancer Institute of New Jersey, New Brunswick, NJ 08903, USA. [10] Institute of Clinical Medicine, University of Oslo, Oslo, Norway. [25] These authors jointly supervised this work: Xavier Tekpli, Vessela N. Kristensen. *A list of authors and their affiliations appears at the end of the paper. ✉email: xavier.tekpli@medisin.uio.no; v.n.kristensen@medisin.uio.no

Transcriptional programs shape cancer cell phenotypes. In breast cancer, clinically relevant subtypes have been identified based on gene expression. Luminal A, Luminal B, Her2-enriched, Basal-like, and Normal-like subtypes have different natural histories, prognosis, and responses to therapies. These subtypes can be identified based on the expression of 50 genes (PAM50)[1].

Luminal A/B tumors are typically estrogen receptor (ER) positive, with Luminal B having a higher expression of proliferation-related genes. Her2-enriched tumors overexpress genes belonging to the ERBB2 pathway, while Basal-like tumors are usually negative for both ER and Her2, and for the progesterone receptor, and to a high degree reflect the triple-negative subgroup[2].

We have previously shown that transcriptional programs between these subtypes are different and underlie their classification[3]. However, phenotypic heterogeneity within each subgroup pertains and could help to further refine subtyping and individualized treatment options.

Long non-coding RNA (lncRNA) expression is highly cell type and tissue specific[4,5]. lncRNAs play important roles in gene regulation, both at the transcriptional and posttranscriptional levels and may help shaping cell type and tissue phenotypes. Several studies have shown that genomic location of lncRNA overlap with enhancer regions[6,7], and that lncRNA promoters may contain chromatin marks associated both with active promoters and enhancers[8].

A substantial number of lncRNAs are tethered to chromatin at or near transcription start sites and can modulate transcription in *cis* through the recruitment of transcription factors (TF) and chromatin modifiers[9,10]. One of the possible effects of lncRNAs in regulating other genes' transcription is through the modulation of enhancer activity and recruitment of proteins that establish and stabilize chromatin conformation[11,12].

In breast cancer, lncRNAs have been implicated in tumor progression, resistance to treatment[13] and in activating the transcriptional network leading to metastasis[14]. Subtype-specific lncRNA expression has previously been described, particularly in the The Cancer Genome Atlas Breast Invasive Carcinoma (TCGA-BRCA) cohort, however, with limited statistical power and validation[13,15,16]. In addition, tumor-specific epigenetic alterations have been identified at lncRNA promoters[15], and lncRNA function has been assessed through pathway enrichment of neighboring genes[16]. Yet the function and regulation of the majority of lncRNAs in breast cancer pathogenesis remains unknown, especially in a subtype-specific manner.

In this study we identify the robust association of lncRNA expression to clinicopathological features in two large population-based cohorts: the Sweden Cancerome Analysis Network – Breast (SCAN-B) initiative and the TCGA-BRCA. Using co-expression analysis of lncRNAs with protein-coding genes, we reveal the cell type-specific expression of lncRNA in breast tumors. To further understand the regulatory network driving lncRNA expression in breast cancer, we combine the clinical and genomic annotation of lncRNA with epigenetic data, transcription factor binding sites, and long-range chromatin interaction information.

## Results

### lncRNA expression according to breast cancer clinicopathological subtypes

To identify lncRNAs expressed by specific breast cancer subtypes or associated with clinicopathological features, we analyzed RNA-sequencing data from two large independent breast cancer cohorts: SCAN-B (n = 3455)[17] and TCGA-BRCA (n = 1095).

We focused on lncRNAs annotated in the Ensembl[18] v93 non-coding reference transcriptome (Supplementary Fig. 1), and

identified 4108 lncRNAs present in both cohorts, which are further analyzed in this study. A small number of lncRNAs (100 in SCAN-B, 37 in TCGA) were expressed >1 transcript per million (TPM) in all patients, but the majority of lncRNAs were expressed at lower levels. The two cohorts differ in the distribution of patients expressing lncRNAs>1TPM (Supplementary Fig. 2). Such sparsity of the lncRNA expression in the two cohorts highlights the importance of analyzing at least two independent breast cancer cohorts to robustly identify the lncRNA associated with clinicopathological features. Hierarchical clustering of the log2 expression values of the 4108 lncRNAs clearly grouped ER positive from ER negative patients, (Fig. 1a: SCAN-B and Fig. 1b: TCGA), indicating an association between breast cancer subtypes and lncRNA expression.

To further identify the lncRNAs associated with breast cancer subtypes, we performed differential expression analysis using linear modeling and empirical Bayes moderation. We report lncRNAs with significant differential expression according to the ER status (Fig. 1c) and HER2 status (Fig. 1d). The significant Pearson correlation between the log fold change (FC) in the SCAN-B and the TCGA cohorts (r = 0.93, p-value < 2.2e-16: ER status and r = 0.75, p-value < 2.2e-16: HER2 status) show that we identify with high confidence lncRNA differentially expressed according to pathological breast cancer subtypes.

On each plot (Fig. 1c, d), we highlight the lncRNAs with the highest absolute fold changes in each breast cancer subgroup. Detailed results from the differential expression analysis are available in Supplementary Data 1. *FOXCUT* was the most significantly deregulated lncRNA over-expressed in ER negative tumors with the highest fold change in both SCAN-B and TCGA, it has been previously shown to enhance proliferation and migration in ER negative breast cancer cell lines[19].

We further performed all pairwise differential expression analyses within the five molecular PAM50 subtypes, Luminal A, Luminal B, Her2-enriched, Basal-like and Normal-like. Figure 1e shows the results of such analysis for Luminal A *versus* Luminal B, two subtypes considered to be closely related, as they are both ER positive; however, we still report 1448 differentially expressed lncRNA between these two subtypes. All pairwise comparisons considering PAM50 subtypes are presented in Supplementary Fig. 3 and Supplementary Data 1.

Few lncRNAs have been associated to patient outcome[20]. To assess the relevance of lncRNA expression robustly and systematically with regards to breast cancer prognosis, we performed Cox proportional hazards regression analysis in the SCAN-B cohort in ER + and ER patients separately. 305 lncRNAs were significantly associated to overall survival of ER + patients in the SCAN-B cohort, of which *MAPT-AS1*, *AP000851.1*, *AP000851.2*, and *ROCR* could be validated in TCGA-BRCA (Supplementary Data 2). *MAPT-AS1* has been previously shown to be associated with better patient outcome in breast cancer patients[21]. *ROCR*, the lncRNA with highest expression in the Luminal A subtype was also associated with ER + prognosis. 77 lncRNAs were associated to overall survival within the ER- group in the SCAN-B cohort, however, none of these were significantly associated with survival after multiple testing correction in the TCGA-BRCA cohort.

To our knowledge, this initial analysis is the first to robustly identify differentially expressed lncRNAs according to breast cancer clinicopathological features and molecular subtypes in two large and independent cohorts.

### Clustering lncRNAs according to high degree of co-expression with protein coding mRNAs

To associate lncRNA expression to known biological functions, we used a co-expression approach

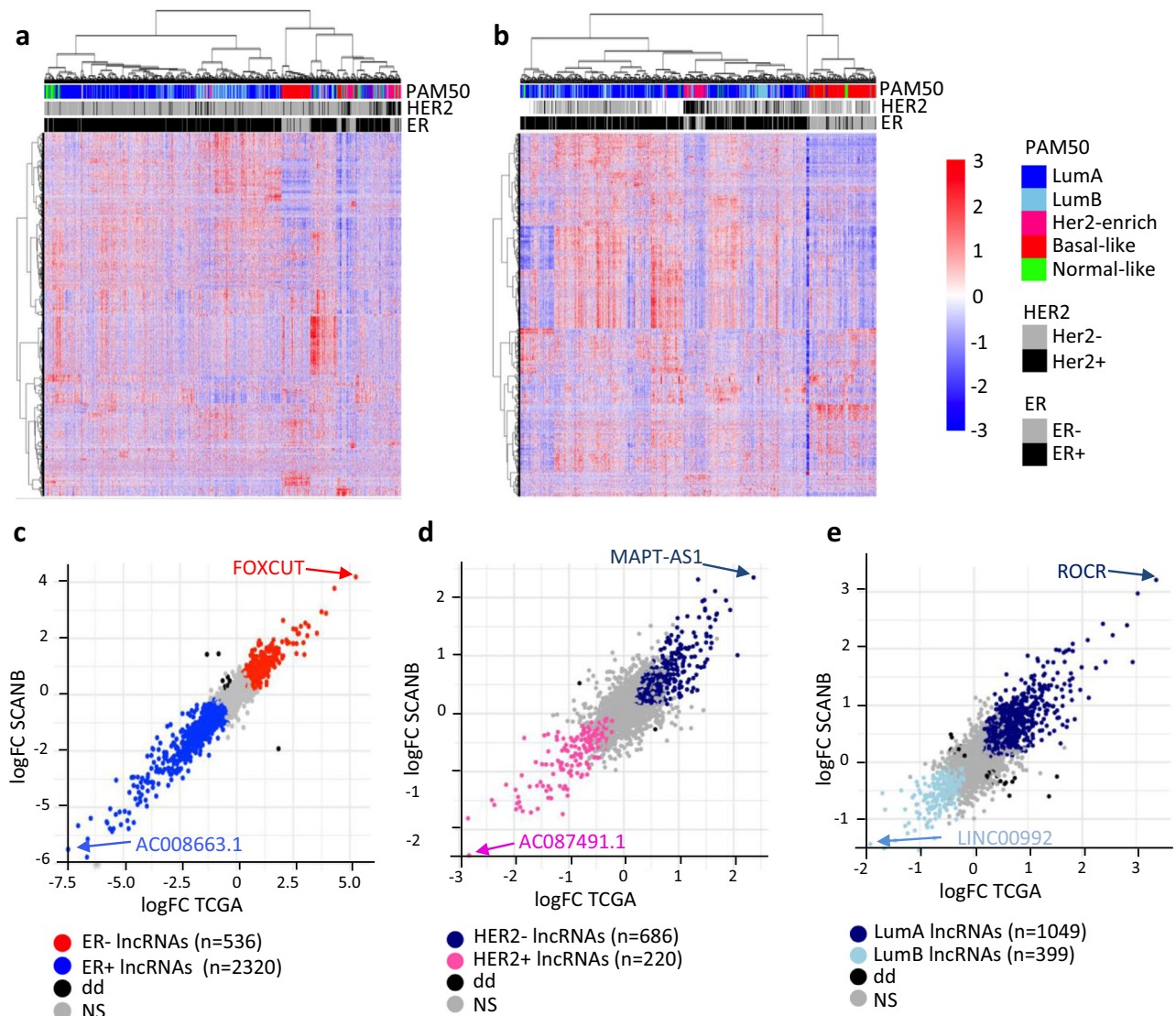

**Fig. 1 lncRNA expression in breast cancer subtypes.** Hierarchical clustering of log2(TPM + 1) of 4108 lncRNAs expressed above filtering thresholds (see Methods) in the SCAN-B (**a**), and TCGA-BRCA (**b**) cohorts. Estrogen Receptor (ER) and Her2 status, as well as PAM50 subtypes are annotated at the top of the heatmap. The expression gradient (blue to red) represents scaled and centered log2(TPM + 1). **c–e** Dot plot of the log Fold Change (FC) from the differential expression analysis using a fitted Limma model (lmfit) and moderated t-statistic (eBayes) between patients of different subtypes in SCAN-B (x-axis) and TCGA-BRCA (y-axis). Each dot represents a lncRNA, while the colour indicates the subtype with the highest expression **c** ER positive (blue) and ER negative (red), **d** Her2 negative (dark blue) and Her2 positive (pink). **e** Luminal A (dark blue), and Luminal B (light blue). Gray dots are lncRNAs that are not significantly differentially expressed, while black dots represent lncRNAs with opposite fold change (FC) in the two cohorts. The number of patients in each clinical group were as follows: ER positive ($n = 2409$ and $n = 807$), ER negative ($n = 504$ and $n = 237$), Her2 positive ($n = 458$ and $n = 114$), Her2 negative ($n = 2845$ and $n = 650$), Luminal A ($n = 1769$ and $n = 562$), and Luminal B ($n = 766$ and $n = 209$) in SCAN-B and TCGA-BRCA respectively.

(Supplementary Fig. 4a) between lncRNA ($n = 4108$) and protein coding genes´ mRNA ($n = 17060$). Retaining the significant Spearman correlation coefficients of all lncRNA-mRNA associations in both cohorts (Bonferroni corrected $p$-value < 0.05), led to $n = 15407856$ significant correlations. On average, each lncRNA was significantly correlated with the expression of 95 mRNAs (Supplementary Fig. 4b), while each mRNA was on average correlated with 20 lncRNAs (Supplementary Fig. 4c). Among the lncRNAs associated to the expression of the highest number of mRNAs, we found a non-coding RNA activated by DNA damage (*NORAD*), known to regulate genome stability[22], as well as other lncRNAs with known function in DNA-damage response, including the estrogen responsive *LINC01488*[23].

We then performed unsupervised clustering of the significant correlations with an absolute Spearman Rho >0.4 and involving

lncRNAs and mRNAs with more than the average number of significant correlations (Supplementary Fig. 4b, c). All significant correlations fulfilling these criteria are denoted in Supplementary Data 3. We identified three lncRNA clusters (x-axis) which correlated with three mRNA clusters (y-axis) (Fig. 2a). Interestingly, most of the correlation coefficients (99.8%) were positive, showing more positive correlations between mRNA and lncRNA than expected by chance. To assess whether the discovery of our three biclusters was driven by the filtering criteria used to select lncRNA and mRNA, an unsupervised clustering including all lncRNAs and mRNAs allowed the rediscovery of the three biclusters, however with much more sparsity (Supplementary Fig. 5).

To link the clustered lncRNAs to the differential expression analysis performed according to breast cancer subtypes, we

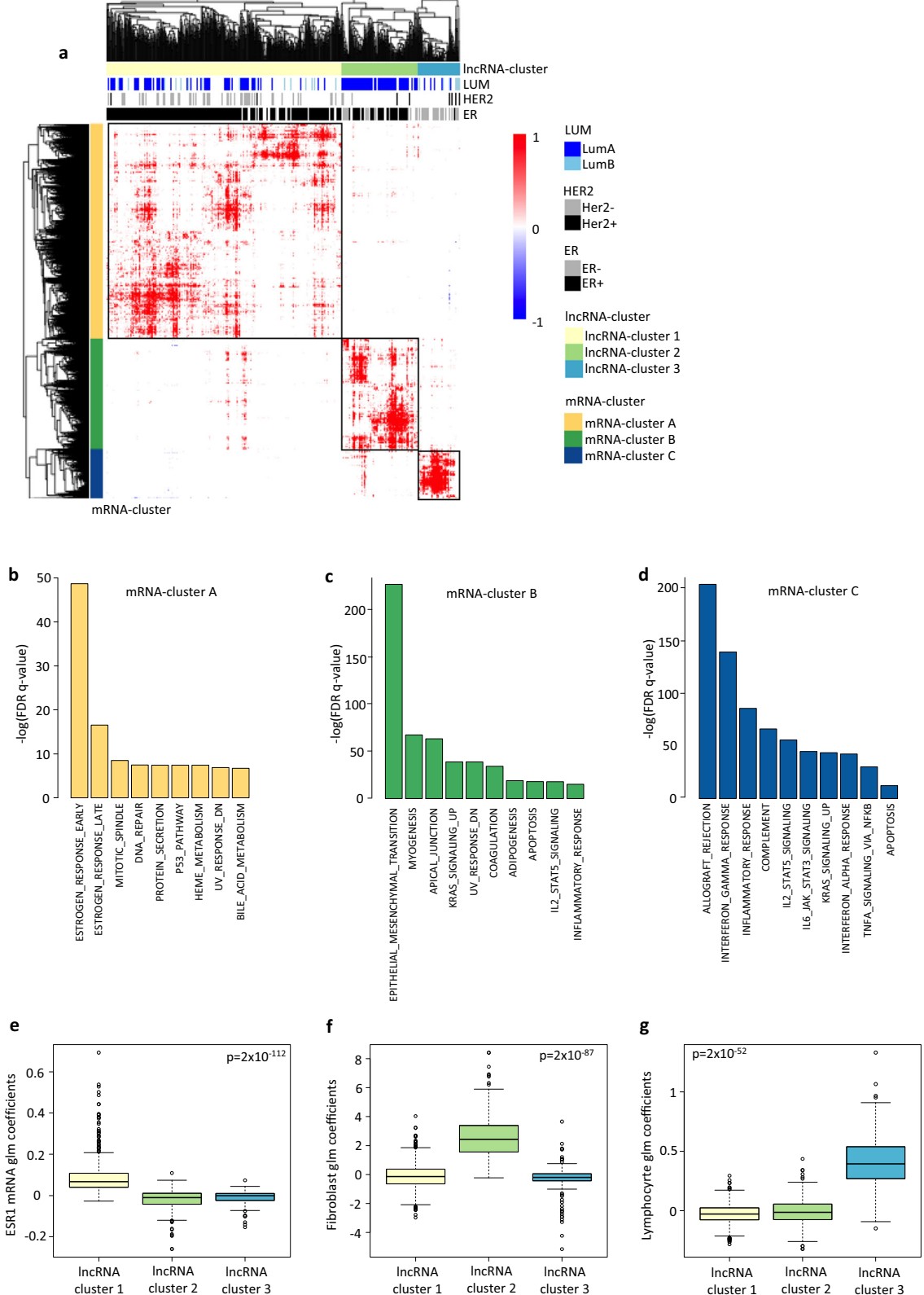

annotated lncRNAs according to whether they were found overexpressed in the respective groups compared in Fig. 1c–e (column annotations of the heatmap). We observed that lncRNA-cluster 1 and 3 were populated by lncRNAs overexpressed in ER positive and ER negative cases, respectively.

**Grouping lncRNAs into pathways related to breast cancer pathogenesis.** Following the unsupervised clustering (Fig. 2a), we found a high degree of significant and dominantly positive correlations between the (i) lncRNAs in cluster 1 and the mRNAs in cluster A, (ii) lncRNA-cluster 2 and mRNA-cluster B and

**Fig. 2 Clustering of lncRNA into relevant pathways for breast cancer. a** Hierarchical clustering of lncRNA-mRNA Spearman correlation values (positive correlation in red, negative correlation in blue) following co-expression analysis between lncRNAs ($n = 4108$) and protein coding mRNAs ($n = 17060$). Only lncRNA and mRNA with significant correlation (Bonferroni $p$-value $< 0.05$) and $-0.4 >$ Spearman's rho $> 0.4$ in the TCGA ($n = 1095$) and SCAN-B ($n = 3455$) cohorts are used in the unsupervised clustering. In addition, we plot only lncRNAs and mRNAs with number of association higher than the mean value of association (Supplementary Fig. 4). Clusters are defined using cutree_rows = 3 and cutree_cols = 3. lncRNAs (x-axis) are annotated according to the differential expression analysis (Fig. 1). **b**, **d** Bar plot showing -log(FDR $q$.value) from a hypergeometric test (y-axis) of gene set enrichment analysis using Hallmark pathways of the MSigDB database. Input genes for GSEA are genes from mRNA-cluster A ($n = 2890$) (**b**), mRNA-cluster B ($n = 1480$) (**c**), and mRNA-cluster C ($n = 667$)(**d**). Boxplot of the coefficients from the generalized linear modeling of the expression of lncRNAs in the SCAN-B cohort using three variables into the same model, *ESR1* mRNA (to reflect estrogen signaling (**e**)), fibroblast score (to infer fibroblast tumor content (**f**)) and lymphocyte score (to infer lymphocyte infiltration (**g**)). Each dot represents the coefficient for a variable and each lncRNA in cluster 1 ($n = 610$), cluster 2 ($n = 199$), and cluster 3 ($n = 110$). Kruskal-Wallis test $p$-values are shown. The line within each box represents the median. Upper and lower edges of each box represent 75th and 25th percentile, respectively. The whiskers represent the lowest datum still within [$1.5 \times$ (75th − 25th percentile)] of the lower quartile, and the highest datum still within [$1.5 \times$ (75th − 25th percentile)] of the upper quartile.

(iii) lncRNA-cluster 3 and mRNA-cluster C. By performing Gene Set Enrichment Analysis (GSEA) using the genes of each mRNA-cluster as input, we could infer by association the pathways the lncRNA-clusters may functionally be associated with.

*lncRNA-cluster 1 & mRNA-cluster A - Estrogen signaling cluster.* 91% of the lncRNAs in cluster 1 were significantly overexpressed in ER positive cases, when compared to ER negative, associating these lncRNAs with estrogen signaling. Further, we found that GSEA analysis of genes in mRNA-cluster A were significantly associated with the estrogen signaling pathway (Fig. 2b, Supplementary Data 3).

*lncRNA cluster 2 & mRNA-cluster B - Fibroblast cluster.* 52% of the lncRNAs in cluster 2 were significantly overexpressed in ER positive and 21% in ER negative cases. Interestingly, 87% were overexpressed in Luminal A tumors when compared to Luminal B.

According to GSEA, genes of mRNA-cluster B are involved in Epithelial to Mesenchymal Transition (EMT) and Apical junctions (Fig. 2c, Supplementary Data 3). There is a high similarity between mesenchymal cells and fibroblasts[24], and fibroblasts are strongly associated with shaping of the extra cellular matrix[25]. In addition, fibroblasts have been shown to be highly abundant in Luminal A breast tumors[26]. We therefore hypothesized that lncRNAs from cluster 2 may be expressed by fibroblasts of the tumor microenvironment.

*lncRNA cluster 3 & mRNA-cluster C - Immune cluster.* For the third lncRNA-cluster, 61% of the lncRNA were overexpressed in ER negative tumors. Protein coding genes of mRNA cluster C were highly correlated with lncRNA-cluster 3 and enriched among pathways reflecting activation of the immune system (Fig. 2d, Supplementary Data 3). Given the fact that ER negative tumors have significantly higher immune infiltration than ER positive tumors[27], we hypothesized that lncRNAs from cluster 3 may be expressed by immune cells and / or modulate the immune tumor microenvironment.

**Predicting cell type expression of lncRNAs.** Having set hypotheses on which pathways and cell types clustered-lncRNA may be associated with, we aimed at providing further evidence for the cell type specific expression of lncRNAs using different approaches.

First, we modeled lncRNA expression as a multivariate function of *ESR1* mRNA expression, fibroblast and lymphocyte infiltration scores reflecting fibroblast or lymphocyte tumor content. We tested which of the three variables explained best each lncRNA's expression (Supplementary Data 4). To infer the lymphocyte and fibroblast content and calculate lymphocyte and

fibroblast scores, we used bulk gene expression and the Nanodissect [23] or xCell[28] algorithms, respectively (see Methods).

We found that *ESR1* expression, fibroblast score, and lymphocyte score were the most significant explanatory variables for 82% of lncRNAs in cluster 1, 60% of lncRNAs in cluster 2 and 84.5% of lncRNA in cluster 3, respectively. Furthermore, when comparing the logistic regression coefficients, which reflect how much each variable explains lncRNA expression, we found that in average the *ESR1*-coefficients were significantly higher in cluster 1 (Fig. 2e, SCAN-B and Supplementary Fig. 6a, TCGA), the fibroblast-coefficients significantly higher in cluster 2 (Fig. 2f, SCAN-B and Supplementary 6b, TCGA) and the lymphocyte coefficient significantly higher in cluster 3 (Fig. 2g, SCAN-B and Supplementary Fig. 6c, TCGA).

These detailed analyses clearly divide lncRNA expressed in breast cancer in three categories, they are either expressed by cancer cells and belong to the estrogen signaling pathways or they are expressed by the main cell types of the tumor microenvironment: lymphocytes and fibroblasts.

**Expression of lncRNAs in breast cancer cell lines and single cell RNA-seq data.** To clearly associate lncRNAs with cell type specific expression in breast cancer, we investigated lncRNA expression in a panel of breast cancer cell lines[29]. Differential expression analysis of breast cancer cell lines according to molecular subtypes confirmed that the lncRNAs with significantly high expression in Luminal A and Luminal B (ER +) cell lines belonged to cluster 1. The top three lncRNAs for both Luminal subtypes are shown in Supplementary Fig. 7a, b. Overall, cluster 1 lncRNAs were expressed at higher levels in the Luminal cell lines (Supplementary Fig. 7c, d). Cluster 2 lncRNAs, which we identify as mainly being expressed in fibroblasts of the tumor microenvironment, showed highest expression in cell lines of the Normal-like subtype. In cluster 3, 20% of the lncRNAs were not expressed in any breast cancer cell lines, but the remaining cluster 3 lncRNAs had the highest expression in Basal, Claudin-low, and Normal-like, ER- cell lines (Supplementary Fig. 7e–h). All the lncRNAs that significantly defined each subtype cell lines from the rest are included in Supplementary Data 5.

We next interrogated single cell RNA sequencing data from a study of 26 breast cancer patients[30]. Following dimensionality reduction and clustering of the 94357 cells from the study by Wu et al., we observed that the cluster of cells obtained overlapped perfectly with the cell type annotation published by the authors[30], which included nine main cell types: normal epithelial, cancer epithelial, myeloid, T-cells, B-cells, endothelial cells, plasmablasts, Cancer Associated Fibroblasts (CAFs) and perivascular-like (PVL)-fibroblasts (Fig. 3a).

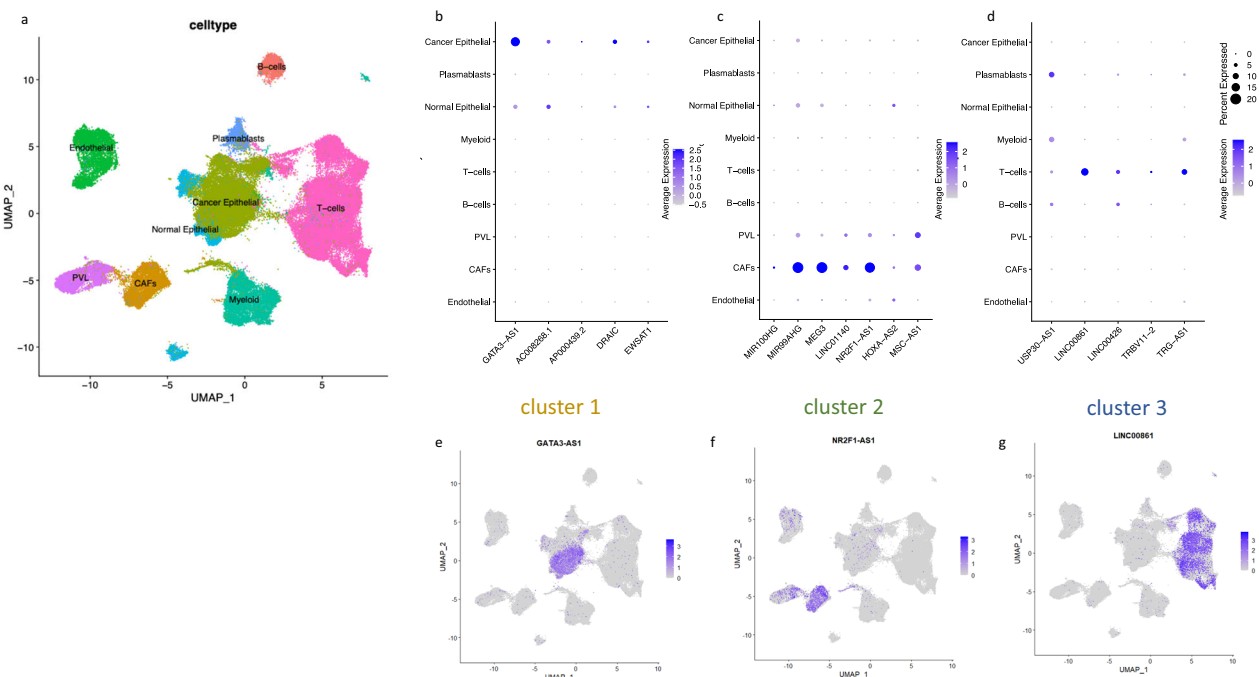

**Fig. 3 lncRNA expression in single cell RNA-seq data. a** UMAP of 94357 single cells from breast tumours colour-coded according to cell types. **b, d** Dot plot of lncRNAs (found in the scRNA-seq data set) with highest glm coefficient associated with the characteristics of each cluster, i.e *ESR1* mRNA (Cluster 1), fibroblast score (Cluster 2), lymphocyte score (Cluster 3). Size of the dot represents the percentage of cells expressing the lncRNA, while the colour of the dot reflects the average expression in each of the UMAP-cell-type-cluster identified. Cluster 1 lncRNAs **b**, cluster 2 lncRNAs **c**, and cluster 3 lncRNAs **d**. **e–g** Expression of one high ranking lncRNA from each lncRNA cluster plotted on the scRNA-seq UMAP. Cluster 1-lncRNA: *GATA3-AS1* **c**, cluster 2-lncRNA: *NR2F1-AS1* **d**, and cluster 3-lncRNA: *LINC0861* **e**. Colour gradient (purple) represents Log normalized counts using scale.factor = 10000.

Dot plot analysis which reflects both average expression and percentage of cells expressing lncRNAs was performed for the lncRNAs with the highest logistic regression coefficient associated with each cluster characteristic feature (i.e, *ESR1* expression for cluster 1, fibroblast infiltration for cluster 2, and immune infiltration for cluster 3) (Fig. 3b, d). We confirmed that lncRNAs of cluster 1 were expressed at higher levels in cancer epithelial cells, cluster 2-lncRNAs were mainly expressed by cancer associated fibroblasts, while lncRNAs of cluster 3 were expressed by immune cells. We further illustrate the expression of *GATA3-AS1* (Fig. 3e), *NR2F1-AS1* (Fig. 3f) and *LINC00861* (Fig. 3g) on a Uniform Manifold Approximation and Projection for Dimension Reduction (UMAP). *LINC00861* has been shown to be expressed in T-Cells in the tissue microenvironment (TME) of lung adenocarcinoma patients and was associated with better prognosis[31]. This lncRNA was also associated with better outcome in ER- patients in the SCAN-B cohort (Supplementary Data 2). Additional illustrations of lncRNA expression on the UMAP are included as Supplementary Fig. 8.

With these analyses, we directly identified lncRNA expression in either breast cancer cells, including cell lines, immune cells, or fibroblast of the breast tumor microenvironment.

**Transcriptional regulation of expression at lncRNA promoters.** lncRNAs are typically co-expressed with protein coding mRNA neighboring genes[4]. We aimed at characterizing lncRNAs regulatory regions in breast cancer.

To focus only on lncRNA specific regulatory regions and avoid analyzing regulatory regions from protein coding genes, we selected lncRNAs for which the promoter regions (transcription start site (TSS) −200/+100 bp) did not overlap with protein coding genes (Fig. 4a, Supplementary Data 6). Indeed, lncRNAs with promoters overlapping with protein coding genes had a

higher level of co-expression with neighboring protein coding genes than independent lncRNAs and the nearest protein coding mRNA (Supplementary Fig. 9). We therefore further analyzed the promoters of lncRNA with no overlap with protein coding gene loci; either promoters of lncRNAs overexpressed in ER positive (n = 2320) or ER negative (n = 536) samples. We compared these two groups of promoters with respect to i) Chromatin accessibility measured by ATAC-seq in 74 TCGA-BRCA patients, ii) ChromHMM, chromatin genome segmentation, and iii) Transcription Factor (TF) - binding sites using the UniBind database[32].

*lncRNA promoters are accessible in an ER-status specific manner.* We found lncRNA promoters to be accessible in a lineage specific manner, *i.e.* promoters of lncRNA overexpressed in ER positive tumors were more open (higher Assay for Transposase-Accessible Chromatin using sequencing (ATAC-seq) signal) in ER positive samples than in ER negative samples. Similarly, promoters of lncRNAs over-expressed in ER negative tumors showed significantly higher ATAC-seq signal in ER negative samples (Fig. 4b, c, Supplementary Data 6), suggesting that lncRNA promoters are highly regulated in a subtype specific manner.

*lncRNA promoters are defined as active regions according to chromHMM.* We assessed whether lncRNA promoters were enriched for specific chromHMM regions defined in subtype specific breast cancer cell lines. We mainly observed significant enrichment for 'Promoter Flanking' and 'Enhancer' (Fig. 4d, e, Supplementary Data 6). When expanding the window upstream of the TSS, the enrichment for 'Enhancer' marks became even more significant, with the lncRNAs over-expressed in ER negative tumors showing particularly significant overlap with 'Enhancer' marks in Basal like cell lines (Supplementary Fig. 10).

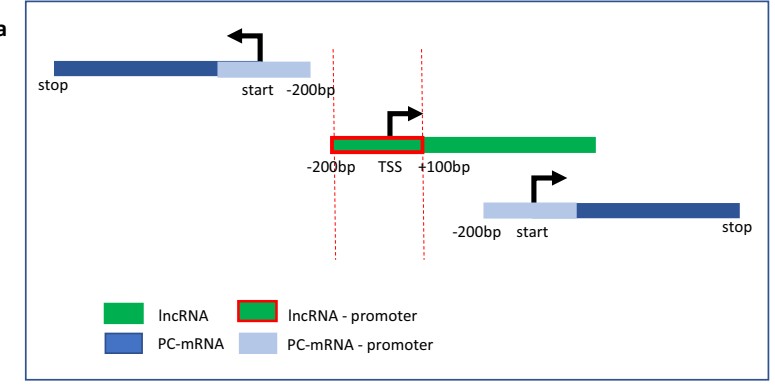

*Specific transcription factors binding sites are found at lncRNA promoters.* We next sought for enrichment in transcription factor binding sites (TFBS) in lncRNAs promoters using the UniBind database[32]. UniBind stores TFBSs with both experimental and computational evidence for direct TF-DNA interactions. We found ER + lncRNA promoters enriched for FOXA1 and ESR1 binding sites; TFs known to drive ER positive breast cancer

(Fig. 4f, Supplementary Data 6). On the other hand, promoters of the lncRNAs highly expressed in ER negative tumors were enriched for BATF3, MAF, and RELA, components of the NF-κB TF complex, shown to be constitutively active in triple negative breast cancer[33] (Fig. 4g, Supplementary Data 6).

To further assess the specificity of the TF binding according to length of the promoter chosen for the lncRNA, we assess three

**Fig. 4 Functional annotation of lncRNA promoters. a** Schematic overview of the definition of lncRNA promoters not overlapping with a protein coding gene locus. bp: base pair; PC: protein-coding; TSS: transcription start site. **b**, **c** Average normalized counts for ATAC-seq peaks mapped to lncRNA promoters in estrogen receptor (ER) positive (+) (blue dots) (n = 58) and ER negative (−) (red dots) (n = 12) breast tumor samples from the TCGA-BRCA cohort. Wilcoxon test p-values are denoted. The line within each box represents the median. Upper and lower edges of each box represent 75th and 25th percentile, respectively. The whiskers represent the lowest datum still within [1.5 × (75th − 25th percentile)] of the lower quartile, and the highest datum still within [1.5 × (75th − 25th percentile)] of the upper quartile. **b** Promoters of independent lncRNAs overexpressed in ER positive cases and **c** promoters of independent lncRNAs overexpressed in ER negative cases. **d**, **e** Enrichment of independent lncRNA promoters across ChromHMM genome segmentation from breast cancer cell lines. Enrichment is calculated as the ratio between the frequency of lncRNA promoters found within a specific segment type, over the frequency of all lncRNA promoters within the same segment type. The length of the bars (x-axis) shows the log transformed BH corrected p-value from the hypergeometric test. **d** Promoters of independent lncRNAs overexpressed in ER positive cases and **e** promoters of independent lncRNAs overexpressed in ER negative cases. Active Enhancer=EhAct, Active Promoter = PrAct, Repeat Zink Finger = RpZNF, Flanking Promoter region = PrFlk. **f**, **g** Swarm plots showing enrichment of TF binding sites (−(log10(p-value) using Fisher's exact tests) on the y-axis for specific sets of promoters according to UniBind. TF names of the top 10 enriched TF binding sites data sets are annotated by colours. **f** Promoters of independent lncRNA overexpressed in ER positive cases and **g** promoters of independent lncRNAs overexpressed in ER negative cases.

different sizes of promoters: TSS −300/+100 bp, TSS −500/ +100 bp, TSS −1000/+100 bp. For ER + lncRNAs binding of ESR1 and FOXA1 dominated for all window sizes (Supplementary Fig. 11a–c). When extending the window upstream of the TSS for ER- lncRNA there was also enrichment for CEBPB, a transcription factor involved in inflammatory response[34], and several additional AP-1 family members with known function in dendritic cell identity[35] (Supplementary Fig. 11d–f).

Altogether, these results gave insight into the regulatory programs specifically at lncRNA promoters and showed that this regulation is closely related to estrogen receptor status in breast cancer.

**Identifying distal regulatory regions for lncRNA.** Finally, we sought for distal regulatory regions for lncRNA in breast cancer. We used our previously published method[36], which is efficient at identifying distal enhancer and long-range interactions between enhancers and promoters through negative correlations between DNA methylation and transcript expression. We correlated the levels of DNA methylation at CpGs and lncRNA expression for all CpGs and lncRNAs on the same chromosome in two cohorts for which DNA methylation and lncRNA expression were available TCGA-BRCA (n = 603) and OSLO2 (n = 279). As the OSLO2 lncRNA expression was measured by Agilent microarray 60 K, we focused on 1027 lncRNAs found in both cohorts (Supplementary Fig. 12). For both cohorts, we identified 26342 CpGs significantly inversely correlated with 396 lncRNA (Bon-ferroni corrected Spearman correlation p-value < 0.05). We first tested in which chromHMM regions the CpGs whose DNA methylation was inversely correlated with lncRNAs were located and found them significantly enriched in enhancer regions (Fig. 5a, Supplementary Data 7). CpGs negatively correlated with lncRNAs highly expressed in ER positive tumors were found in open chromatin regions significantly more open in ER positive samples according to the TCGA-BRCA ATAC-seq data (Fig. 5b). Correspondingly, CpGs negatively correlated with lncRNAs highly expressed in ER negative breast cancer were found in regions significantly more open in ER negative tumors (Fig. 5c). Further confirming that the CpG in cis inverse correlation with lncRNA expression pointed at biologically relevant and active distal regulatory regions, we found such CpGs near binding sites of TFs described at breast cancer enhancers (Fig. 5d).

The *LINC01488* locus provides a good illustration of distal regulatory regions possibly involved in the regulation of lncRNA expression (Fig. 5e). *LINC01488* expression showed negative correlation to distant CpGs on the same chromosome in the TCGA-BRCA and OSLO2 cohorts (Fig. 5e). A specific negative correlation between *LINC01488* expression and DNA methylation levels at a CpG (cg00211115) in an upstream active enhancer

region is shown in Fig. 5f (OSLO2) and Fig. 5g (TCGA). This CpG has lower levels of methylation in ER positive patients and was found to reside within the binding sites of key transcription factors (ESR1, FOXA1, and GATA3, ChIP-seq). Furthermore, experimental long-range interactions defined by Pol2 binding (ChIA-PET Pol2 data), showed an interaction, loop, between the distal enhancer and *LINC01488* TSS (Fig. 5e). *LINC01488* was also detected in a long-range interaction with *CCND1* (Fig. 5h) and showed significant correlation to *CCND1* expression in both SCAN-B (Fig. 5i) and the TCGA-BRCA cohort (Supplementary Fig. 13). Other examples of lncRNAs with inverse correlation with DNA CpG methylation at enhancer sites that reside in long-range interactions are shown in Supplementary Figs. 13 and 14 and Supplementary Data 7, or lncRNAs in long-range interactions with protein coding mRNAs (Supplementary Data 7).

Altogether, these analyses show that integration of lncRNA expression with DNA methylation and long-range interaction data aids in identifying subtype-specific distal regulatory regions for lncRNA.

## Discussion

This study is, to our knowledge, the first to identify lncRNAs associated to clinicopathological features in breast cancer using two large independent cohorts. Combining the analysis of the SCAN-B and TCGA-BRCA RNA-seq data allowed us to assess the expression of more than 4000 lncRNAs with respect to breast cancer clinicopathological features and to report lncRNAs with robust association to clinical features across patient cohorts with remarkable concordance. We identified more than 2800 lncRNA genes, almost 70% of all the lncRNAs included in this analysis, with significant differential expression between ER positive and ER negative breast cancer. This is in line with the previous observations based on the TCGA-BRCA cohort alone[13].

Characterization of lncRNA functions remains a critical challenge[37]. Here, to approach this question from an in silico point of view, we grouped lncRNAs based on their correlation with all protein coding mRNAs using hierarchical clustering. This expands on previous studies that have focused on selected lncRNAs or identified pathway enrichment of neighboring mRNAs[15,16]. Our analysis revealed how lncRNA expression is related to underlying features of inter- and intra-tumor hetero-geneity. While lncRNAs of cluster 1 were mainly over-expressed in ER positive breast cancers and were found to be associated with estrogen signaling, the expression of lncRNAs of cluster 2 and 3 were mainly explained by cells from the tumor micro-environment. This further underlines the highly cell- and tissue-specific expression of lncRNAs[4,5].

Cluster 2 – lncRNAs had their expression mostly explained by an in silico computed fibroblast score. We further verified this

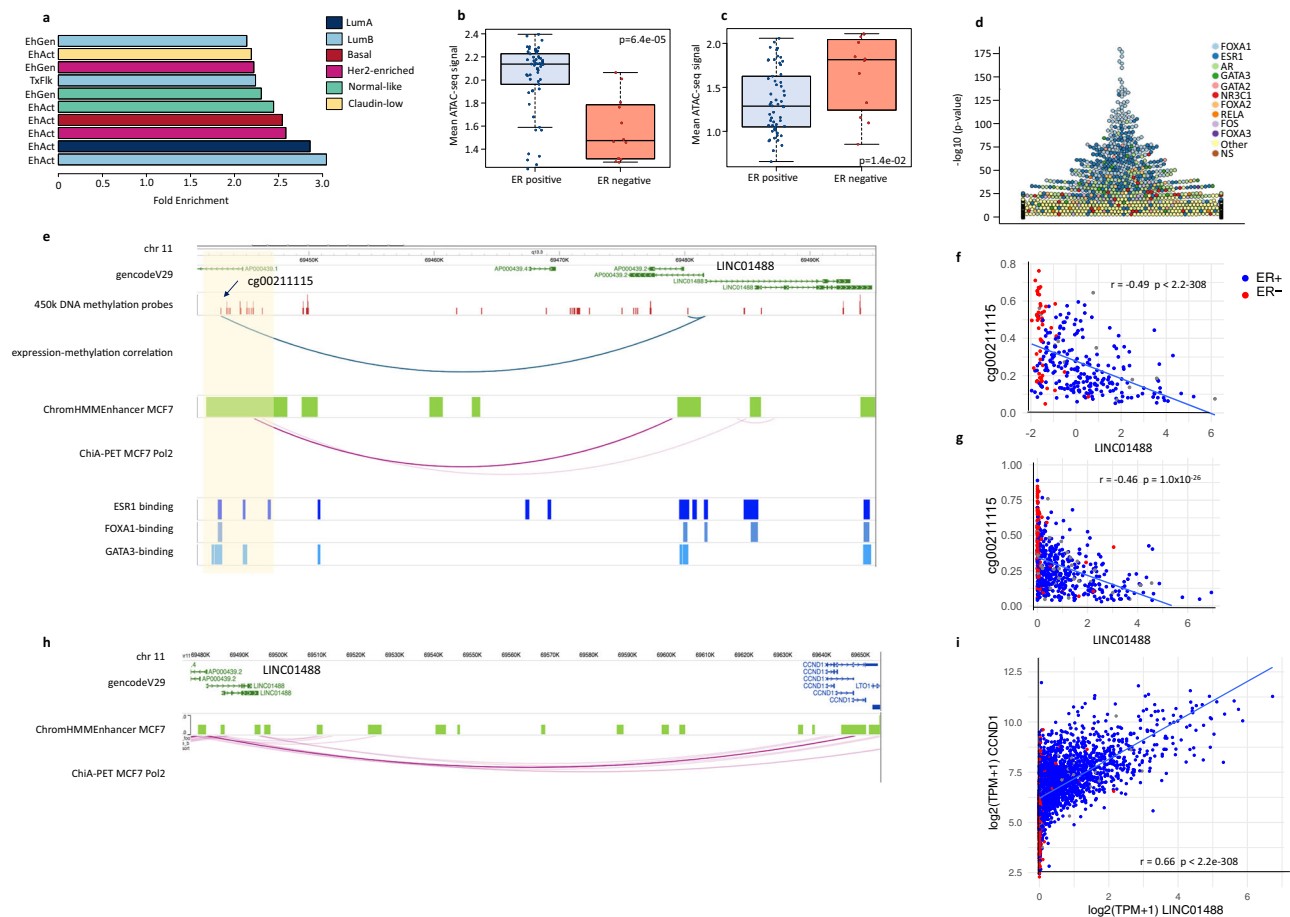

**Fig. 5 Distal regulatory element in the *LINC01488* locus. a** Enrichment of CpGs with DNA methylation significantly inversely correlated with lncRNA expression across ChromHMM genome segmentation from breast cancer cell lines. Enrichment is calculated by comparing the genomic location of the CpG inversely correlated to all the CpGs on the 450k Illumina array as background. Active Enhancer = EhAct, Ehnancer Genic = EhGen, Transcription flanking = TxFlk. Average normalized counts for ATAC-seq peaks mapped to CpG location for which DNA methylation is significantly inversely correlated with lncRNAs with higher expression in ER positive cases (**b**) and higher expression in ER negative cases (**c**). ATAC-seq data from ER + (blue dots) (n = 58) and ER- (red dots) (n = 12) breast tumor samples from the TCGA-BRCA cohort. Wilcoxon test p-values are denoted. The line within each box represents the median. Upper and lower edges of each box represent 75th and 25th percentile, respectively. The whiskers represent the lowest datum still within [1.5 × (75th − 25th percentile)] of the lower quartile, and the highest datum still within [1.5 × (75th − 25th percentile)] of the upper quartile. **d** Swarm plot showing enrichment of TF binding sites (−(log10(p-value) using Fisher's exact tests) on the y-axis for CpGs with DNA methylation inversely correlated with lncRNA expression. Names of the top 10 enriched TF binding sites data sets are annotated by colours. **e** Graphical illustration of the *LINC01488* locus annotated for different epigenomic tracks. CpGs measured by the 450 k Illumina array are shown together with the significant negative correlations between levels of DNA methylation and *LINC01488* expression in the OSLO2 and TCGA cohorts (blue arcs, negative expression-methylation correlation). ChromHMM Enhancer regions (active and genic) in the Mcf7 cell line (green) with ChiA-PET polII loop connecting the TSS of *LINC01488* to the CpG in the enhancer region (pink arcs). TF binding of ESR1 (dark blue), FOXA1 (blue), and GATA3 (light blue) from ChIP-seq experiments (ReMap). **f, g** Correlation plot of levels of *LINC01488* expression (x-axis) and levels of DNA methylation of the CpG (y-axis) in long-range interaction in **e**. Rho and p-value from Spearman correlation is indicated. **f** OSLO2 (ER positive, n = 214, ER negative, n = 52), **g** TCGA (ER positive, n = 807, ER negative, n = 237). **h** Graphical illustration of the *LINC01488* locus annotated with ChromHMM Enhancer regions (active and genic) in the Mcf7 cell line (green) and ChiA-PET polII loop connecting *LINC01488* to *CCND1* (pink arcs). **i** Correlation plot of log2(TPM + 1) *LINC01488* expression (x-axis) and log2(TPM + 1) *CCND1* expression (y-axis) in ER positive (n = 2409) and ER negative (n = 504) patients in the SCAN-B cohort. Rho and p-value from Spearman correlation are indicated.

observation using single cell RNA-seq data and confirmed that many cluster 2 -lncRNAs were expressed by fibroblasts. One such lncRNA was *MEG3*, which was shown to contribute to the development of cardiac fibrosis[38]. Further, the expression of the mature miRNAs hsa-miR-99a and hsa-miR-100 have previously been associated to fibroblasts in breast cancer[39]. We found that the corresponding miRNA precursor transcripts which themselves are lncRNAs were part of cluster 2 and were indeed expressed in fibroblasts of the tumor breast microenvironment. Pathway enrichment of the cluster 2 associated mRNAs showed association to EMT. Expression of a cluster 2 lncRNA, *ROCR*, has

been reported to regulate SOX9 expression in both mesenchymal stem cells[40], and basal-like breast cancer cells, where it promoted proliferation[41]. In this study we identify *ROCR* as the lncRNA that most significantly differentiates Luminal A and Luminal B breast cancer patients. Interestingly, *NR2F1-AS1* has recently been shown to be up-regulated in mesenchymal-like breast cancer stem-like cells, contributing to tumor dissemination[42]. Here, we show clear expression of this lncRNA in CAFs, and higher expression in the Luminal A subtype. Crosstalk between CAFs in the tumor microenvironment and cancer cells can regulate epithelial to mesenchymal transition (EMT) markers and promote

invasion and metastasis[43], and further studies are needed to establish whether other lncRNAs from cluster 2 directly contribute to invasiveness in breast cancer.

Gene set enrichment terms for mRNAs associated to cluster 3 lncRNAs point to hot tumors with high immune infiltration. We were able to identify several lncRNAs from this cluster which were expressed by tumor infiltrating immune cells. A recent pan-cancer study of patients in the TCGA cohort identified a panel of immune-related lncRNAs [34] which could stratify non-small cell lung cancer in three subgroups with differences in response to chemotherapy, and prognosis. Cluster 3 lncRNAs were identified as regulators of immune-related pathways in[44] and had higher expression in ER negative patients. Knowledge about the specific cell types that express lncRNAs can improve our understanding of their function in cancer. We believe our identification of immune-related and fibroblast-associated lncRNAs can serve as a useful resource to choose relevant model systems for more in-depth functional characterization of lncRNAs.

To identify transcriptional regulation of lncRNAs, independent of the regulation of the neighbor protein coding gene, we first separated lncRNA based on whether their promoters overlap with the protein coding gene loci or not. lncRNA-mRNA pairs where the lncRNA promoters were located within the protein coding gene locus showed significantly higher correlation than other lncRNA-mRNA pairs. This has been reported previously in AML patients and cell-lines and indicates a shared *cis*-regulation between lncRNAs and protein coding gene at the same locus[45].

Higher enhancer activity has been attributed to lncRNA transcription[7], and tissue-specific expression of lncRNAs at enhancer regions suggests a role in determining lineage-specific gene expression [4]. We found an enrichment of chromatin features associated with active enhancer regions in lncRNA promoters, which may further indicate that these lncRNAs originate from subtype specific regulatory elements that are active in cancer cells.

The most significant enrichment at lncRNA promoters with high expression in ER- patients was for repeat sequences/ZNF gene clusters. Repeat and transposable elements play a role in both the origin, and regulation of lncRNAs [40], and we cannot rule out transcription in these areas due to hypo-methylation/de-repression of otherwise silenced genomic regions.

The transcription factors FOXA1 and ESR1 bind to active enhancers in breast cancer[46,47], and are important for lineage determination[36]. We found enrichment for FOXA1 and ESR1 binding sites at the independent promoters of lncRNAs with high expression in ER positive patients, which provides further evidence for an association between the expression of some of these lncRNAs to enhancer function.

lncRNAs with enhancer functions can regulate nearby protein coding genes [36]. *LINC01488* has been shown to mediate breast cancer risk by playing a role in homologous recombination (HR)-mediated DNA repair. The risk SNP resides in a distant enhancer of *CCND1*, which is also involved in estrogen induction of *LINC01488* expression[23]. Here, we identify several distal enhancer regions in long-range interaction with the TSS of *LINC01488*. We show lower levels of DNA methylation at these enhancers in ER positive patients. The lncRNA is also in long-range interaction with the neighboring gene, *CCND1*. *LINC01488* shares a bivalent promoter with *AP000439.2* This lncRNA was not detected in the OSLO2 cohort (measured by microarray), and it is possible that the same distal regulatory regions are involved in regulation of both these neighbor lncRNAs.

A significant correlation between *LINC01488* and *CCND1* expression was observed in both the SCAN-B and TCGA-BRCA cohorts. In the study by Betts et al. knockdown of *LINC01488* resulted in decreased expression of *CCND1*. Further studies are necessary to determine the role of *LINC01488* on *CCND1* expression, and to identify other enhancer lncRNAs that may function in gene regulation of protein coding genes in breast cancer subtypes.

In conclusion, we find a large number of lncRNAs with specific expression related to clinicopathological features in breast cancer. In breast cancer lncRNA expression associate to specific pathways known to play a role in pathogenesis, as well as specific cell types infiltrating breast tumors. We show that promoters of lncRNAs are enriched in regulatory regions and TF relevant to breast cancer, indicating active transcriptional regulation and association to lineage specific enhancers in breast cancer subtypes.

## Methods

**Patient material**. Two independent breast cancer cohorts with RNA-seq data were used; SCAN-B ($n = 3455$)[48] and The Cancer Genome Atlas Breast Invasive Carcinoma (TCGA-BRCA) cohort ($n = 1095$)[49]. A third independent cohort, the OSLO2 breast cancer cohort for which lncRNA expression were measured by Agilent 60 K array[50,51], was also included.

*SCAN-B cohort*. The SCAN-B cohort[17,48] is a consecutive observational cohort of resectable primary breast cancers from south Sweden. Patients included in this study were enrolled in the Sweden Cancerome Analysis Network - Breast (SCAN-B) initiative (ClinicalTrials.gov ID NCT02306096), approved by the Regional Ethical Review Board in Lund, Sweden (Registration numbers 2009/658, 2010/383, 2012/58, and 2013/459). All patients provided written informed consent prior to study inclusion. All analyses were performed in accordance with patient consent and ethical regulations and decisions. Patient characteristics and clinicopathological features are described in[17], and are according to current clinical definitions in Sweden. 3455 patients were identified with high quality RNA sequencing (RNA-seq) data and included in this analysis with the following clinical groups: ER positive ($n = 2409$), ER negative ($n = 504$), Her2 positive ($n = 458$), Her2 negative ($n = 2845$), Basal like ($n = 341$), Luminal A ($n = 1769$), Luminal B ($n = 766$), Her2 ($n = 310$), and Normal-like ($n = 206$) (Supplementary Data 8). RNA-seq library preparation and sequencing methods are described in[48]. Quantification of gene expression was performed using *kallisto*[52] (v0.46.0) with 100 bootstrap samples (−b 100), using an indexed reference that combined all Ensembl[18] coding and non-coding sequences (Homo_sapiens.GRCh38.cdna.all.fa and Homo_sapiens.GRCh38.ncrna.fa, Ensembl Archive Release 93 (July 2018)). Transcript abundance from *kallisto* were summarized to gene level expression using tximport[53] (v1.16.1) in R. lncRNAs were defined as genes in the Ensembl (v93) non-coding reference with a length above 200 bp. lncRNA expressed at >1 TPM in >5% of samples in the cohort (Supplementary Fig. 1), and with an interquartile range >0.1 (IQR function in R) were included in the downstream analysis (Supplementary Data 9). Hierarchical clustering of patients was performed using *hclust* as part of the pheatmap package (v1.0.12) in R with correlation distance and ward D2 as agglomeration method (Fig. 1a, b).

*The Cancer Genome Atlas Breast Invasive Carcinoma (TCGA-BRCA) cohort*. The Cancer Genome Atlas Breast Invasive Carcinoma (TCGA-BRCA) cohort, from here on named TCGA, has previously been described[49]. Clinical information for the TCGA was obtained from the UCSC Xena browser[54] (https://tcga.xenahubs.net/download/TCGA.BRCA.sampleMap/BRCA_clinicalMatrix, curated survival end-points; https://tcga-pancan-atlas-hub.s3.us-east-1.amazonaws.com/download/Survival_SupplementalTable_S1_20171025_xena_sp; Full metadata), and PAM50 subtype information from[55] were obtained using the TCGAbiolinks package in R[56] (Version:2.16.3). After removing formalin-fixed, paraffin-embedded (FFPE) and duplicate samples 1095 patients were included in the analysis with the following clinical groups: ER positive ($n = 807$), ER negative ($n = 237$), Her2 positive ($n = 114$), Her2 negative ($n = 650$), Basal like ($n = 190$), Luminal A ($n = 562$), Luminal B ($n = 209$), Her2 ($n = 82$), and Normal-like ($n = 40$) (Supplementary Data 8). To quantify lncRNA and protein coding gene expression, raw fastq files from the TCGA BRCA cohort were downloaded from https://gdc.cancer.gov/. Sample identifiers and clinical information is included in Supplementary Data 8. Quantification of gene expression was performed as described for SCAN-B with tximport v 1.10.1, and same filtering was applied as described above (Supplementary Data 10).

DNA methylation data from TCGA[55](level 3), probes with more than 50% missing values were removed, and further missing values were imputed using the function pamr.knnimpute (R package pamr) with k = 10.

*The Oslo2 breast cancer cohort*. The Oslo2 breast cancer cohort has been previously described[39,50,51] and is a consecutive study collecting material from breast cancer patients with primary operable disease at several hospitals in south-eastern Norway. Patients were included in the years 2006–2019. The study was approved by the Norwegian Regional Committee for Medical Research Ethics (approval number 1.2006.1607, amendment 1.2007.1125), and patients have given written informed

consent for the use of material for research purposes. All experimental methods performed are in compliance with the Helsinki Declaration. The mRNA expression data have been previously published and can be obtained from GEO with accession number GSE58215[50]. To accurately assign array probes to lncRNAs, published probe sequences (GEO Platform GPL14550) were aligned to Ensembl.93 non-coding reference sequences (Homo_sapiens.GRCh38.ncrna.fa) using blast (ncbi-blast-2.6.0). Probes where all 60 bp matched 100% to the reference were included in the analysis. In the case where several probes could detect the same lncRNA, the mean expression value was used. A total of 4018 probes mapping to 3000 unique Ensembl gene IDs were included in the lncRNA analysis, and 1027 lncRNAs were detected in all three cohorts according to the filtering criteria described for TCGA and SCAN-B.

The Illumina Infinium HumanMethylation450 microarray was used to measure the DNA methylation levels (GSE84207)[57,58]. Preprocessing and normalization involved steps of probe filtering, color bias correction, background subtraction and subset quantile normalization. The DNA methylation data have been previously published[36].

**Differential gene expression analysis**. "scaledTPM" values from the tximport function were used to create a DGEList object using edgeR (v 3.24.3 (TCGA)/3.30.3 (SCAN-B)), and linear modeling (lmFit) and the empirical Bayes moderation function (eBayes) from the Limma/voom R-package (v 3.38.3/3.44.3) were used to define differentially expressed lncRNAs in the TCGA and SCAN-B cohorts. lncRNAs with Benjamini-Hochberg adjusted[59] p-values < 0.05 were considered significant, and lncRNAs that were significant in both cohorts were included in the downstream analysis. lncRNAs referred to as ER + and ER− associated had higher expression in the respective clinical group in both cohorts (Supplementary Data 1).

**Survival analysis**. Cox proportional hazards regression analysis was performed using the coxph function of the Survival package (v3.3-1) in R with Overall Survival as endpoint. ER + and ER− patients were analyzed separately in the SCAN-B cohort and lncRNAs with p-value < 0.05 after Benjamini-Hochberg (FDR) correction were used for validation in the TCGA BRCA cohort (Supplementary Data 2).

**Correlation to protein coding genes expression and hierarchical clustering of lncRNAs**. Log2(TPM + 1) expression values for lncRNAs were correlated to all protein coding genes with an interquartile range >0.1 (IQR function in R) in the TCGA and SCAN-B cohorts, using Spearman correlation (cor.test in R). lncRNA-mRNA pairs with p-value < 0.05 after Bonferroni correction[59] in both cohorts were included in the subsequent analysis (Supplementary Data 9 and 10). lncRNAs and mRNAs were filtered prior to clustering, retaining only those with i) Spearman Correlation coefficient below −0.4 and above 0.4 in both cohorts, and ii) more than the average number of associations (n = 95 lncRNAs, n = 20 mRNAs, Supplementary Fig. 3b, c). For clustering, Spearman correlation values were binarized to −1/1 for negative and positive correlation respectively. Hierarchical clustering was performed using hclust as part of the pheatmap package (v1.0.12) in R with correlation distance and average linkage. To identify and decide upon the number of lncRNA and mRNA clusters, the dendrograms were visually inspected using different cut-offs on the cutree_rows and cutree_cols functions of the pheatmap package. Cut-offs were manually selected to define the clusters depicted in Fig. 2a (cutree_rows = 3 and cutree_cols = 3).

**Gene Set Enrichment analysis (GSEA)**. Gene Set Enrichment Analysis was carried out using either the 50 Hallmark pathway gene sets[60] (h.all.v7.0.symbols.gmt), or "C5", ontology gene sets[61,62] (c5.all.v7.0.symbols.gmt) from MSigDB[63]. Enrichment was calculated using hypergeometric test (the R function phyper) of the mRNAs in each cluster, against all genes in a gene set. P-values were FDR corrected, and the top 10 pathways with adjusted p-value < 0.05 were used.

**Lymphocyte and fibroblast infiltration scores**. The Nanodissect algorithm[64] (http://nano.princeton.edu/) was used for in silico estimation of lymphocyte infiltration. The breast collection data (May 2013), which contains 17940 genes measured on 622 arrays, was inspected for genes specifically expressed in lymphocytes (standard genes; n = 476; available online and defined from expert literature review) and not expressed in mammary gland (n = 777) or mammary epithelium (n = 79). The genes with more than 65% probability to be positive lymphocyte-specific standard genes as opposed to mammary gland or epithelium were further used in downstream analysis to score each SCAN-B and TCGA-BRCA samples for the level of lymphocyte infiltration. The average expression of the set of standard genes in a sample reflected lymphocyte infiltration. The xCell algorithm[28] was used to obtain a fibroblast score for SCAN-B samples with log2 (TPM + 1) values as input. For TCGA, xCell scores were downloaded from https://xcell.ucsf.edu/xCell_TCGA_RSEM.txt.

**lncRNA expression modeled with generalized linear models**. Generalized linear modeling (glm function in R) was used to model lncRNA expression as a function of ESR1 mRNA expression, fibroblast infiltration, and lymphocyte infiltration to estimate which variable(s) explained most each lncRNA expression. Resulting coefficient of such modelling are used in subsequent analysis to estimate the impact of each variable in lncRNA clusters.

**RNA-seq from breast cancer cell lines**. Gene expression from cell lines representing different breast cancer molecular subtypes: MCF7 and ZR751 (luminal A), MB361 and UACC812 (luminal B), AU565 and SKBR3 (HER2), MB469 and HCC1937 (basal), MB231 and MB436 (Claudin-low), and MCF10A and 76NF2V (Normal breast), each 4 replicates (GSE96860[29],) was obtained from the Recount3 project[65] (v 1.2.6) using the recount::getTPM function in R. 911 of the 919 lncRNAs defined in the clustering analysis (Fig. 2a) were available and used to identify differentially expressed lncRNAs in each subtype compared to all other subtypes (wilcox test) using the FindAllMarkers function of the Seurat package (v4.1.0) in R.

**Single cell RNA-seq from breast cancer patients**. Count matrix of single cell RNA-seq[30] were analyzed using the Seurat package (v3.2.1) in R to obtain UMAP. In brief, count matrix were already filtered for dying cells by the authors. It was further normalized and scaled regressing out potential confounding factors (number of UMIs, number of gene detected in cell. percentage of mitochondrial RNA). After scaling, variably expressed genes were used to construct principal components (PCs) and PCs covering the highest variance in the dataset were selected based on elbow and Jackstraw plots to build the UMAP. Clusters were calculated by the FindClusters function with a resolution between 0.8 and 1.8, and visualized using the UMAP dimensional reduction method.

Nine main cell types were identified on these UMAP based on the authors annotations. The main cell types identified are normal epithelial, cancer epithelial, myeloid, T, B, endothelial cells, plasmablasts, CAF and perivascular-like -fibroblasts.

**lncRNA promoter annotation**. lncRNA promoters were defined as Transcription Start Site (transcription_start_site), positions obtained from Ensembl (v.93) using BioMart[66] (biomaRt_2.45.6, host = 'http://Jul2018.archive.ensembl.org') −200 bp (upstream of TSS) and +100 bp downstream, and by increasing the upstream window to −300, −500, and −1000. lncRNA transcripts with independent promoters were obtained using bedtools subtract, with the -A flag, of all lncRNA promoters from a background file containing a window spanning 200 bp (300 bp, 500 bp, and 1000 bp for expanding window sizes) upstream of protein coding gene start positions, to gene end position (BEDtools[67], v2.29.2), remaining transcripts were regarded as overlapping promoters. Overlapping protein coding genes were identified using the bedtools intersect command with the same input as described above, and nearest protein coding gene to independent lncRNAs were identified by bedtools nearest using the default parameters with lncRNA promoter regions (−200/ + 100) and protein coding genes start-stop coordinates.

**ATAC-seq data from TCGA-BRCA**. Normalized ATAC-seq peak signals (log2((count + 5)PM)−qn) for 74 TCGA breast tumors[68] were downloaded from the Xena browser[54] (https://xenabrowser.net/datapages/). lncRNA promoter positions (−200/ + 100) were intersected with the peak positions using bedtools intersect. To test for differential open regions between ER positive and negative tumors, the average normalized counts of the peaks containing lncRNA promoters were calculated per tumor sample and a Wilcoxon rank-sum test was applied to test for statistical significance using R. lncRNA promoters associated to ER + or ER- tumors were tested separately.

**Enrichment of ChromHMM regions at lncRNA promoters**. For functional annotation of the lncRNA promoters, we utilized the ChromHMM segmentation from Xi et al.[29]. obtained from cell lines representing different breast cancer molecular subtypes: MCF7 and ZR751 (luminal A), MB361 and UACC812 (luminal B), AU565 and SKBR3 (HER2), MB469 and HCC1937 (basal), MB231 and MB436 (Claudin-low), and MCF10A and 76NF2V (Normal breast). These segmentations were derived from ChIP-seq data for five histone modification marks (H3K4me3, H3K4me1, H3K27me3, H3K9me3, and H3K36me3) to predict thirteen distinct chromatin states: active promoters (PrAct) and promoter flanking regions (PrFlk), active enhancers in intergenic regions (EhAct) and genic regions (EhGen), active transcription units (TxAct) and their flanking regions (TxFlk), strong (RepPC) and weak (WkREP) repressive polycomb domains, poised bivalent promoters (PrBiv) and bivalent enhancers (EhBiv), repeats/ZNF gene clusters (RpZNF), heterochromatin (Htchr), and quiescent/low signal regions (QsLow). We intersected the lncRNA promoters, window sizes as described above, (hg19 coordinates obtained with the UCSC liftOver tool, https://genome.ucsc.edu/cgi-bin/hgLiftOver) with the segmented genomes from the cell lines (BEDtools intersect) and assessed enrichment of lncRNA promoters with different clinical association (DE analysis, ER + and ER- lncRNAs Supplementary Data 1), within each of the 13 chromatin states using hypergeometric tests (the R function phyper) with all lncRNA promoters as background (n = 34595). ChromHMM features were filtered to exclude features supported by <10 lncRNA promoters, and p-values were corrected using the Benjamini-Hochberg (BH) procedure[59].

**Enrichment of transcription factors binding sites at lncRNA promoters**. To assess the enrichment of TFBSs at lncRNA promoters (300 bp window described above), we considered the direct TF-DNA interactions (i.e. TFBSs) stored in the updated version of the UniBind database as of 20.10.2020[32]. These TFBSs were obtained by combining both experimental (through ChIP-seq) and computational (through position weight matrices from JASPAR[69]) evidence of direct TF-DNA interactions (see ref. [32] for more details). Note that a TF can have multiple sets of TFBSs derived from different ChIP-seq experiments. The enrichment of UniBind TFBS sets in regions surrounding lncRNA promoters against a universe considering all lncRNA promoter regions (window sizes as described above, Ensembl.93) with the UniBind enrichment tool (https://unibind.uio.no/enrichment/, source code available at https://bitbucket.org/CBGR/unibind_enrichment/; input R data with TFBS information available on zenodo at https://doi.org/10.5281/zenodo.4452896). Specifically, the enrichment is computed using the LOLA R package (version 1.12.0)[70] using Fisher's exact tests. Fig. 4 f and g, and Fig. 5d plot the Fisher's exact p-values using swarm plots (swarmplot function of the seaborn Python package, https://doi.org/10.5281/zenodo.824567) with annotations for the TFs associated with top 10 most enriched TFBS sets.

**DNA methylation lncRNA expression correlation analysis**. Within each data set (OSLO2 and TCGA), CpGs with an interquartile range (IQR) > 0.1 were selected. Considering only CpGs and lncRNAs present in both data sets resulted in 143 631 CpGs and 1027 lncRNAs, and analysis was restricted to lncRNAs and CpGs on the same chromosome (total number of tests $n = 7130824$). To test the correlation between the level of DNA methylation of CpGs and lncRNA expression (log2 expression (OSLO2) or log2 (TPM + 1) (TCGA)), the Spearman correlation statistics was applied (function cor.test with method = "spearman" in R). An association was considered statistically significant if a Bonferroni-corrected p-value was <0.05. Only significant correlations with the same direction (sign) were kept.

We assessed enrichment of all CpGs with negative correlation to lncRNA expression to each of the 13 chromatin states described above using hypergeometric tests (the R function phyper) with all Illumina Infinium HumanMethylation450 BeadChip CpGs as background ($n = 436\,506$). P-values were corrected using the Benjamini-Hochberg (BH) procedure[59].

**ChIA-PET Pol2 data and ChIP-seq peaks**. ChIA-PET Pol2 loop data from the MCF7 cell line was retrieved from ENCODE, accession number ENCSR000CAA[71]. We investigated overlaps between ChIA-PET Pol2 loops and CpGs with negative correlation to lncRNA expression (Supplementary Data 6). A CpG-lncRNA pair was considered to be in a ChIA-PET loop if the CpG and the lncRNA TSS were found in two different feet of the same loop. lncRNA TSS positions were lifted to hg19 coordinates using the UCSC liftOver tool before intersecting with the loop coordinates using BEDtools intersect. Similarly, lncRNA-mRNA pairs were considered to be in a loop if the lncRNA (gene body coordinates) and mRNA (gene body coordinates) were found in two different feet of the same loop. For the specific analyses of MCF7 TF ChIP-seq data sets, we retrieved ENCODE ChIP-seq peak regions from the ReMap 2018[72] database (ENCSR000BST.GATA3.MCF7, ERP000783.ESR1.MCF7, and GSE72249.FOXA1.MCF7).

**Statistics and reproducibility**. All analyses were performed in the R software (4.1.1). The number of patients in each clinical group in the two patient cohorts were as follows: ER positive ($n = 2409$ and $n = 807$), ER negative ($n = 504$ and $n = 237$), Her2 positive ($n = 458$ and $n = 114$), Her2 negative ($n = 2845$ and $n = 650$), Luminal A ($n = 1769$ and $n = 562$), and Luminal B ($n = 766$ and $n = 209$) in SCAN-B and TCGA-BRCA respectively. Linear modeling (lmFit) and the empirical Bayes moderation function (eBayes) from the Limma/voom R-package (v 3.38.3/3.44.3) were used to define differentially expressed lncRNAs. Benjamini-Hochberg adjusted p-values < 0.05 were considered significant for all tests, unless otherwise stated. Cox proportional hazards regression analysis was performed using the coxph function of the Survival package (v3.3-1) in R with Overall Survival as endpoint. lncRNA ($n = 4108$)-mRNA ($n = 17060$) and lncRNA ($n = 1027$)-CpG ($n = 143631$) methylation correlation analysis was performed using Spearman correlation (cor.test in R) and Bonferroni corrected for multiple testing. Hypergeometric tests (the R function phyper) were used for GSEA of mRNA clusters (mRNA-cluster A ($n = 2890$), mRNA-cluster B ($n = 1480$), and mRNA-cluster C ($n = 667$)), as well as ChromHMM enrichment analysis. Generalized linear modeling (glm function in R) was used to model lncRNA expression as a function of ESR1 mRNA expression, fibroblast infiltration, and lymphocyte infiltration to estimate which variable(s) explained most each lncRNA expression in SCAN-B ($n = 3455$) and TCGA ($n = 980$). Kruskal-Wallis test was used to assess glm coefficients from the three lncRNA Clusters, cluster 1 ($n = 610$), cluster 2 ($n = 199$), and cluster 3 ($n = 110$). Difference in ATAC-seq signal from ER+ ($n = 58$) and ER-($n = 12$) breast tumor samples from the TCGA-BRCA cohort was evaluated by Wilcoxon rank-sum test. Fisher's exact tests were used to calculate enrichment of TF binding sites.

Individual statistical tests are described in the relevant sections above and in figure legends.

**Reporting summary**. Further information on research design is available in the Nature Research Reporting Summary linked to this article.

## Data availability

Clinical information for the TCGA is available from the UCSC Xena browser [54] (https://tcga.xenahubs.net/download/TCGA.BRCA.sampleMap/BRCA_clinicalMatrix, curated survival endpoints; https://tcga-pancan-atlas-hub.s3.us-east-1.amazonaws.com/download/Survival_SupplementalTable_S1_20171025_xena_sp; Full metadata), and PAM50 subtypes can be obtained using the TCGAbiolinks package in R[56]. Clinical information for the SCAN-B cohort is available from[17]. Clinical annotation for samples used throughout this manuscript is available in Supplementary Data 8. Source data underlying Fig. 1a is presented in Supp Data 9, Fig. 1b in Supp Data 10, and Fig. 1c–e in Supp Data 1. Source data underlying Fig. 2a–d is presented in Supp Data 3, and data underlying Fig. 2e–g is available in Supp Data 4 and 8 (Lymphocyte and Fibroblast scores). Gene expression from breast cancer cell lines is available through the Recount3 project[65] (GSE96860[29]). The Count matrix of single cell RNA-seq used in Fig. 3 can be obtained from [30]. Normalized ATAC-seq data used for Figs. 4b, c, and 5b, c can be accessed through the Xena browser[54] (https://xenabrowser.net/datapages/). ChromHMM segmentation data from breast cancer cell lines used for Figs. 4d, e and 5a were obtained from Xi et al.[28]. TF-DNA interactions used for Fig. 4 f, g are available from the UniBind database at https://unibind2018.uio.no (29). Source data for Fig. 4 is presented in Supp Data 6. Clinical data including ER status and lncRNA expression data from the OSLO2 breast cancer cohort can be obtained from GEO with accession number GSE58215 and DNA methylation data is available at GEO with the accession number GSE84207. The sample key to combine GSE58215 (gene expression) and GSE84207 (DNA methylation) for the OSLO2 patient cohort is available upon request. ChIA-PET (ENCODE) and TF ChIP-seq (ReMap) data from MCF7, can be used for Fig. 5e, h, can be obtained from ENCODE (ENCSR000CAA; https://www.encodeproject.org/experiments/)[33], and ReMap 2018[72] database (ENCSR000BST.GATA3.MCF7, ERP000783.ESR1.MCF7, and GSE72249.FOXA1.MCF7). Source data for Fig. 5 is available in Supplementary Data 7. The authors declare that the main data supporting the findings of this study are available within the article and its Supplementary Data files.

## Code availability

No custom code was used to generate data used in this study. R packages and specific functions, as well as software used are described in relevant sections in the method section.

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

## Acknowledgements

S.B. and M.R.A. are postdoctoral fellow funded by the Norwegian cancer Society (grant nr 190372-2017). The authors would like to acknowledge patients, clinicians, and hospital staff participating in the SCAN-B study, the staff at the central SCAN-B laboratory at Division of Oncology, Lund University, the Swedish national breast cancer quality registry (NKBC), Regional Cancer Center South, and the South Swedish Breast Cancer Group (SSBCG). This work was supported by the Mrs. Berta Kamprad Foundation, the Mats Paulsson Foundation, the Biltema Foundation, the Swedish Cancer Foundation, Governmental Funding of Clinical Research within National Health Service.

## Author contributions

Conception and design: V.N.K., X.T., G.B., and S.G. Collection and assembly of data: S.B., J.V-C., J.H., S.K., T.F., J.T., X.T., K.K.S., OSBREAC. Data analysis and interpretation: S.B., M.R.A., J.H., S.K., K.B.E., A.M., X.T., V.N.K. Manuscript writing: All authors. Final approval of manuscript: All authors.

## Competing interests

The authors declare no competing interests.

## Additional information

## OSBREAC

Tone F. Bathen[11], Elin Borgen[12], Anne-Lise Børresen-Dale[2,13], Olav Engebråten[6,13,14], Britt Fritzman[15], Olaf Johan Hartmann-Johnsen[15], Øystein Garred[12], Jürgen Geisler[13,16], Gry Aarum Geitvik[2], Solveig Hofvind[17,18], Rolf Kåresen[13,19], Anita Langerød[2], Ole Christian Lingjærde[13,20,21], Gunhild Mari Mælandsmo[6,13,22], Bjørn Naume[13,14], Hege G. Russnes[2,12,13], Torill Sauer[13,23], Helle Kristine Skjerven[24], Ellen Schlichting[19] & Therese Sørlie[2,13]

[11]Department of Circulation and Medical Imaging, Faculty of Medicine and Health Sciences, Norwegian University of Science and Technology (NTNU), Trondheim, Norway. [12]Department of Pathology, Division of Diagnostics and Intervention, Oslo University Hospital, Oslo, Norway. [13]Institute for Clinical Medicine, University of Oslo, Oslo, Norway. [14]Department of Oncology, Oslo University Hospital, 0379 Oslo, Norway. [15]Department of Surgery, Østfold Hospital, Fredrikstad, Norway. [16]Department of Oncology, Akershus University Hospital, Lørenskog, Norway. [17]Cancer Registry of Norway, Oslo, Norway. [18]Department of Health and Care Sciences, UiT – The Artic University of Norway, Tromsø, Norway. [19]Department of Breast- and Endocrine Surgery, Division of Surgery, Cancer and Transplantation, Oslo University Hospital, Oslo, Norway. [20]Centre for Cancer Biomedicine, University of Oslo, Oslo, Norway. [21]Department of Computer Science, University of Oslo, Oslo, Norway. [22]Department of Pharmacy, Faculty of Health Sciences, University of Tromsø, Tromsø, Norway. [23]Department of Pathology, Akershus University Hospital, Lørenskog, Norway. [24]Breast and Endocrine Surgery, Department of Breast and Endocrine Surgery, Vestre Viken Hospital Trust, Drammen, Norway.

