## [Peer Review File · Communications Biology]

Reviewers' comments:

Reviewer #1 (Remarks to the Author):

The authors present a comprehensive computational study of lncRNA expression in breast cancer using two datasets (TCGA and SCAN-B). The findings are overall interesting and contribute to our current understanding of lncRNA biology in cancer. In particular, the use of two highly powered cohorts and the co-expression analyses are of high scientific quality.

However, the second half of the manuscript does not contain many novel insights, with some missed opportunity, such as combining the co-expression analysis with the additionally analyzed features such as long-range chromatin interactions.

There are a number of suggestions for additional analyses that would substantially elevate the manuscript to a publication of high impact and broad scientific interest, as detailed below.

Introduction:

- Not all basal-like tumors are TNBC – the authors might want to clarify and cite accordingly
- A number of previous studies have characterised lncRNAs in different BC subtypes. These studies should be cited and differences between previous studies and this one should be stated more clearly upfront. Early studies of note include for example the 2014 paper by Su et al in *Oncotarget* and the 2015 paper by Yan et al in *Cancer Cell*, who both correlated lncRNA expression with PAM50 classifications using TCGA data. Differences between the findings in this manuscript and previously published studies should also be included in the Discussion

Results:

- Figure 1a: The heatmaps are not very clear in terms of obvious clusters, which may not be overly surprising as some lncRNAs will only be expressed in a small percentage of patients, and some lncRNAs will be ubiquitously expressed. One wouldn't necessarily expect correlation of ER status with all >4,000 lncRNAs here. Sub-grouping might provide a clearer picture, and clarification of how many lncRNAs are expressed across all patients might be helpful.
- I am wondering if some bias towards "hub" genes was introduced in the co-expression analysis by filtering for only those lncRNAs with above average co-expression partners.
- For the lncRNA 2-mRNA B cluster, my first thought regarding the "EMT" and "junction" signature was around invasiveness and metastasis, rather than about tumour-associated fibroblasts. Could this cluster be associated with more invasive types of cancer instead/also (according to subtypes – unlikely given the ones stated, and/or patient data/outcomes for example)? I do appreciate that the correlations look convincing but am curious if the invasiveness hypothesis was explored further in any way. How are "normal" fibroblasts different in their expression compared to CAFs?
- The lncRNA 3 – mRNA C cluster could also be classified as immune response / inflammatory cluster given the GSEA terms point towards inflammation
- Figure 3: It seems that there is surprisingly little overlap between cancer cells and the three clusters in the single cell analysis, and cluster 1 does not seem well represented anywhere. How can this be explained? Is the data analysis or the underlying datasets biased towards the tumor microenvironment rather than actual cancer cells?
- Figure 4: The promoter regions are defined rather narrowly as -200/+10. Do the results change when promoters are defined more conservatively (e.g. TSSs >1 kb apart), or better yet, defined by a guided approach based on epigenetic signatures (where available) and/or actual transcription?
- Figure 4: It is not very surprising that higher expression levels are associated with more open chromatin. It might be more interesting to see whether the chromatin accessibility of co-expressed mRNAs can be predicted by lncRNA expression.
- Similarly, the significance of overlapping chromatinHMMs is unclear. It is well known that many lncRNAs originate from enhancer regions. The authors may want to clarify the significance of the analyses in Figure 4. Not many "new insights" seem to have been gained here.
- Could the reduced number of overexpressed lncRNAs in ER- compared to ER+ BC be one reason on why less significant hits were obtained in the TF analysis? Would expanding/redefining promoters as suggested above be helpful here?
- Figure 4+5: The ChromHMM figures need more detailed descriptions/explanations in the text, currently mostly unclear.

- Figure 5: This seems to be a missed opportunity to bring together the co-expression analysis in this paper and the emerging paradigm of lncRNAs defining TADs. Instead of identifying long-range enhancers of lncRNAs, which does not seem to be highly informative, it would be exciting to see whether there are long-range interactions between any given overexpressed lncRNAs in a particular subtype and its co-expressed mRNAs (and that these long-range interactions are not present in subtypes where the respective lncRNA is not overexpressed). This could provide a mechanism of subtype specific coordination of gene expression via lncRNAs.

- Figure 5: The provided example is confusing, as GATA3-AS overlaps completely with at least one isoform of GATA3, and overall GATA3-AS seems to share a (divergent) promoter with most GATA3 isoforms. Therefore, the method may simply identify long-range interactions of the GATA3 promoter rather than the lncRNA promoter (or rather, the two promoters are the same/overlap). It is unclear and not mentioned how these two possibilities could be distinguished conclusively. An example of an intergenic lncRNA would be a lot more convincing. It would also be good to show more than one example (additional examples can be in the supplement).

Discussion:

- The discussion seems disproportionately long, with some more detailed description of results likely better suited in the Results section of the manuscript.

Reviewer #2 (Remarks to the Author):

The manuscript entitled 'Subtype and cell type specific expression of lncRNAs in breast cancer' by Sunniva Bjørklund et al. is a research article which addresses the associations between lncRNA expression and breast cancer clinicopathological features in two large population-based cohorts (SCAN-B) and the TCGA-BRCA cohort. They discovered three distinct clusters of lncRNAs using co-expression analysis of lncRNAs with protein coding genes. Then they identified the subtype specific transcription factor (TF) occupancy at their promoters, and integrated lncRNA expression with DNA methylation data to identify long range regulatory regions for lncRNA which were validated using ChiA-Pet-Pol2 loops. This research is novel and interesting. However, the following concerns need to be addressed.

Major concerns:

1 The author discovered the difference of lncRNA expression in different subtypes of breast cancer cells, but the author did not further explore the potential significance and validation of this result (clinical significance, auxiliary pathological typing judgment, guidance for treatment, prediction of patient prognosis, etc.). I think the authors should at least detect these lncRNAs expression levels in different kinds of breast cancer cell lines to confirm that these novel lncRNAs are indeed the subtype and cell type specific expression lncRNAs in breast cancer.

2 The authors use the SCAN-B and TCGA-BRCA database to analysis the lncRNAs. And they found several lncRNAs specific expression in a certain subtype of breast cancer, such as FOXUT, MAPT-AS1, LINC00992. Several reports have demonstrated these lncRNAs functions in breast cancer. For instance, it has been reported that MAPT-AS1 is overexpressed in breast cancer but not in triple negative breast cancer (TNBC), and that high expression of MAPT-AS1 is correlated with better patient survival (Biochem Cell Biol. 2019 Apr;97(2):158-164. doi:10.1139/bcb-2018-0039.). The authors should cite these published papers to confirm the finding of different expression of lncRNAs in different subtypes of breast cancer. It will make the authors' bioinformatics analysis results much more solid.

Other minor concerns should be addressed:

1 The data sample information has reached 1097 in TCGA-BRCA of UCSC Xena browser (update time: 2021, March 31th). I suggest the authors update to the newest TCGA-BRCA database to analysis the lncRNAs expression.

2 In figure 1c-d, the authors search for the overlap lncRNAs in the SCAN-B database and TCGA-BRCA database, while we notice that the ER/Her2 positive and negative samples which the authors showed in the two panels are much less than the two databases sample number (SCAN-B n=3455; TCGA-BRCA n=1095). So the authors need state the data cleanout standard in the two databases. Besides, it will be better if the authors can analysis the lncRNAs expression status in the samples both have ER and Her2 information.

Dear editor,

In a cover letter, we have summarized the major analyses that were performed in the lab to address the concerns of the reviewers. Below are the detailed answers:

We hope you will find the referees' comments useful as you decide how to proceed. Should further experimental data or analysis allow you to address these criticisms, we would be happy to look at a substantially revised manuscript. However, please bear in mind that we will be reluctant to approach the referees again in the absence of major revisions.

In particular, please note that the following revisions would be necessary for us to contact our referees again:

1) Perform additional bioinformatics analyses as requested by both reviewers

Done

2) Revise the discussion section to incorporate the answers to the questions from Reviewer 1

Done

3) Perform additional analyses in cell lines, as asked by Reviewer 2 regarding Figure 3.

Done

We are committed to providing a fair and constructive peer-review process. Do not hesitate to contact us if you wish to discuss the revision or if there are specific requests from the reviewers that you believe are technically impossible or unlikely to yield a meaningful outcome.

Reviewer #1 (Remarks to the Author):

The authors present a comprehensive computational study of lncRNA expression in breast cancer using two datasets (TCGA and SCAN-B). The findings are overall interesting and contribute to our current understanding of lncRNA biology in cancer. In particular, the use of two highly powered cohorts and the co-expression analyses are of high scientific quality.

However, the second half of the manuscript does not contain many novel insights, with some missed opportunity, such as combining the co-expression analysis with the additionally analyzed features such as long-range chromatin interactions. There are a number of suggestions for additional analyses that would substantially elevate the manuscript to a publication of high impact and broad scientific interest, as detailed below.

Introduction:

- Not all basal-like tumors are TNBC – the authors might want to clarify and cite accordingly

We agree with the reviewer, a Reference pointing to the distribution of TNBCs in the Basal like subtype has been added (page 3, line 85).

- A number of previous studies have characterised lncRNAs in different BC

subtypes. These studies should be cited and differences between previous studies and this one should be stated more clearly upfront. Early studies of note include for example the 2014 paper by Su et al in *Oncotarget* and the 2015 paper by Yan et al in *Cancer Cell*, who both correlated lncRNA expression with PAM50 classifications using TCGA data. Differences between the findings in this manuscript and previously published studies should also be included in the Discussion

We thank the reviewer for pointing out these studies, we have added the references to the introduction (page 4, lines 108-110), and discussed them (page 15, lines 416-418). Both studies have indeed showed that many lncRNAs are associated with breast cancer subtypes in the TCGA-BRCA cohort. Here we analyze lncRNA expression in an independent, highly powered Breast Cancer cohort (SCAN-B), which allows us to validate some of the findings of these two previous studies, but also make some important new robust discoveries. We would also like to point out that our analyzes of the lncRNA-mRNA co-expression networks allows the identification of lncRNAs expressed in specific cell types of the tumor microenvironment, decomposing the expression from bulk tumor.

Results:

- Figure 1a: The heatmaps are not very clear in terms of obvious clusters, which may not be overly surprising as some lncRNAs will only be expressed in a small percentage of patients, and some lncRNAs will be ubiquitously expressed. One wouldn't necessarily expect correlation of ER status with all >4,000 lncRNAs here. Sub-grouping might provide a clearer picture, and clarification of how many lncRNAs are expressed across all patients might be helpful.

There is indeed only few lncRNA expressed across all patients. To address this comment, we have now included density plots showing the fraction of patients expressing each lncRNA > 1 TPM in both cohorts. A small number of lncRNAs (100 in SCAN-B, 37 in TCGA) are expressed at this level or above across all patients. The density plots are included as Supplementary Figure S2 and have been described in the results on page 5 lines 130-136.

We could not observe that the most abundant lncRNAs contribute to better separation of clinical subgroups. Many of these lncRNAs are involved in housekeeping functions not necessarily related to the clinical presentation of breast cancer. In fact, the most distinct clusters were observed when taking the lncRNAs with the highest variance. Hierarchical clustering of the most variable lncRNAs as well as the 100 lncRNAs expressed above 1 TPM in all patients of the SCAN-B cohort are included below for the use of the Reviewer.

100 lncRNAs TPM>1 in all SCAN-B patients

lncRNAs IQR>1 in SCAN-B

- I am wondering if some bias towards “hub” genes was introduced in the co-expression analysis by filtering for only those lncRNAs with above average co-expression partners.

We agree with the reviewer’s comment that by keeping only lncRNA with an above average significant correlation with an mRNA we may push our analyses to more hub networks. To assess this agnostically, we added a heatmap without filtering as a new Supplementary Figure S5. This heatmap constitutes of ~12000 mRNA, almost half the protein coding genes in the genome, and shows that most mRNAs and lncRNAs have very few associations (white space). We also added to this new clustering the annotation from our main Figure 2a. We could clearly observe that the initially discovered clusters are recapitulated when no filtering based on number of associations was performed. These new analyses led us to conclude that the filtering contributes to obtaining robust clustering involving the hub lncRNA associated with the pathways and cell types linked to the clusters. We have included these new analyses in the result section page 7 lines 198-201.

- For the lncRNA 2-mRNA B cluster, my first thought regarding the “EMT” and “junction” signature was around invasiveness and metastasis, rather than about tumour-associated fibroblasts. Could this cluster be associated with more invasive types of cancer instead/also (according to subtypes – unlikely given the ones stated, and/or patient data/outcomes for example)? I do appreciate that the correlations look convincing but am curious if the invasiveness hypothesis was explored further in any way. How are “normal” fibroblasts different in their expression compared to CAFs?

This comment is pertinent as the role of EMT in invasiveness and metastasis is well described. However, few tumors have metastatic related outcome in the TCGA

cohort and there are no metastatic events registered in the SCAN-B cohort. It was therefore not possible to formally and directly address the relationship between lncRNA and invasiveness / metastasis in these two cohorts.

What we can say from our single cell RNAseq data is that lncRNA from cluster 2 were mainly expressed by fibroblasts. A link between fibroblasts in the TME and invasiveness has been shown before, we are therefore very interested to follow up on the reviewer's comment in other suitable cohorts and future studies. We have added a section in the discussion on page 14 and 15, lines 433-444.

- The lncRNA 3 – mRNA C cluster could also be classified as immune response / inflammatory cluster given the GSEA terms point towards inflammation

We do agree with the reviewer, these gene set enrichments terms point to “hot” tumor with high immune infiltration and higher expression of immune-related lncRNA expression. To address this, we have added this point to the discussion (page 16, lines 444-447).

- Figure 3: It seems that there is surprisingly little overlap between cancer cells and the three clusters in the single cell analysis, and cluster 1 does not seem well represented anywhere. How can this be explained? Is the data analysis or the underlying datasets biased towards the tumor microenvironment rather than actual cancer cells?

We agree with the reviewer. Our explanation was that the lncRNAs in cluster 1 are strongly associated with ER+ tumors, and the single cell dataset used at submission contained few ER+ patients. Together with the low sensitivity of single cell methods to identify lowly expressed genes, this may explain the fact that we did not see much expression of these lncRNAs in the first set of single cell data.

A bigger scRNA-seq dataset, which includes more ER+ breast cancer patients, has in the meantime become available to us: <https://pubmed.ncbi.nlm.nih.gov/34493872/> This allowed a more accurate exploration of lncRNA in different cell types. We found that lncRNAs from cluster 1 with high GLM coefficient to the ESR1 receptor were indeed enriched in the cancer epithelial cells as shown in the new Figure 3a and 3e, and Supplementary Figure S8a-d.

Another reason for the little overlap was that our goal was not to exhaustively report all lncRNAs at a single cell level, but to show that the ones we claim are expressed in certain cell types are indeed expressed where we expected. In this respect we analyze lncRNAs with highest GLM coefficient instead of the lncRNAs that correlate with the highest number of mRNAs. The same applies to Cluster2 and Cluster3 lncRNAs, which were enriched in the fibroblast/CAFs and immune cells, respectively. These new analyses have led us to revise Figure 3 and the related result section (pages 10 and 11, lines 282-306).

In this version of the manuscript, we also performed analysis of breast cancer cell lines, which corroborate the above observations by showing particularly high expression of cluster1 lncRNAs in Luminal A and Luminal B cell lines (Supplementary Figure S7).

- Figure 4: The promoter regions are defined rather narrowly as -200/+10. Do the results change when promoters are defined more conservatively (e.g. TSSs >1 kb

apart), or better yet, defined by a guided approach based on epigenetic signatures (where available) and/or actual transcription?

We agree with the reviewer that the chosen window size of 300 bp is a conservative definition of the promoter region. The Fantom consortium has defined promoters through CAGE analysis and found that the highest peaks (CAGE expression) are in a window from -300 upstream to 100 downstream of the TSS (PMID: 25723102). Therefore, and in agreement with the reviewer's suggestion, we added analysis of expanding window sizes up to 1kb upstream of the TSS. Particularly subtype specific '*Enhancer*' marks become more significant with a wider window, and we have added these results as Supplementary Figure S10, and revised the manuscript page 12, lines 335-338, as well as the discussion page 17, lines 468-471.

- Figure 4: It is not very surprising that higher expression levels are associated with more open chromatin. It might be more interesting to see whether the chromatin accessibility of co-expressed mRNAs can be predicted by lncRNA expression.

We found the reviewer comment interesting.

Our study shows that in cis lncRNA are tightly correlated with surrounding mRNA and chromatin accessibility, it would therefore to our opinion be possible to build models which could predict lncRNA expression based on mRNA expression +/- chromatin accessibility and vice versa. In this case, such models could be used to input lncRNA expression as such transcripts are lowly expressed, we however think that such algorithm and prediction are for the moment out of the scope of the manuscript, but we need to team up with collaborators experts in statistical models for such analyses.

Our goal in using the ATAC-seq data in breast cancer was to understand whether lncRNA which were classified as independent (non-overlapping with protein coding genes) used their own promoters which seems to be the case when finding both open chromatin and relevant transcription factor binding in the vicinity of these independent lncRNA. We are now emphasizing this point further in the revised version of the manuscript (page 11, lines 311-313, page 16, lines 458-460, p 17, lines 468-471)

- Similarly, the significance of overlapping chromatinHMMs is unclear. It is well known that many lncRNAs originate from enhancer regions. The authors may want to clarify the significance of the analyses in Figure 4. Not many "new insights" seem to have been gained here.

We agree with the reviewer that it is not new insight that many lncRNA may originate from enhancers, but rather a confirmation that these lncRNAs originate from bona fide regulatory elements that are active in cancer cells in a subtype specific manner. We added a comment about this page 12, lines 338-339, and page 17, lines 472-473.

- Could the reduced number of overexpressed lncRNAs in ER- compared to ER+ BC be one reason on why less significant hits were obtained in the TF analysis? Would expanding/redefining promoters as suggested above be helpful here?

Indeed, the lower numbers of ER negative promoters inputted into the TF binding analyses could hinder the discovery of subtype specific TF associated with ER negative-lncRNA expression.

As suggested by the reviewer, we increased the regions in which TF binding enrichment was assessed. We found enrichment of more TF in ER- extended promoter region, while the results were mainly unchanged for ER+ promoters. These results are now shown in Supplementary Fig S11.

- Figure 4+5: The ChromHMM figures need more detailed descriptions/explanations in the text, currently mostly unclear.

We added more details to the ChromHMM Figures according to the reviewer request.

- Figure 5: This seems to be a missed opportunity to bring together the co-expression analysis in this paper and the emerging paradigm of lncRNAs defining TADs. Instead of identifying long-range enhancers of lncRNAs, which does not seem to be highly informative, it would be exciting to see whether there are long-range interactions between any given overexpressed lncRNAs in a particular subtype and its co-expressed mRNAs (and that these long-range interactions are not present in subtypes where the respective lncRNA is not overexpressed). This could provide a mechanism of subtype specific coordination of gene expression via lncRNAs.

Indeed, we identify lncRNAs-mRNAs in *pol2* loops in the MCF7 cell line and have included this as a Supplementary Table T8. We have included one such example in Figure 5 (see also answer below). However, to perform more in-depth subtype analyses one is limited by the availability of subtype specific long-range interactions. When we compared the long-range interaction available for an ER- breast cancer cell line (MDAMB-231 PMID: 29731168, PMID: 27643841) to those in ER+-MCF7, we found them very similar, possibly pointing to the fact that the depth of sequencing in ChiA-Pet data is not sufficient to identify subtype specific enhancer-promoter interactions.

- Figure 5: The provided example is confusing, as GATA3-AS overlaps completely with at least one isoform of GATA3, and overall GATA3-AS seems to share a (divergent) promoter with most GATA3 isoforms. Therefore, the method may simply identify long-range interactions of the GATA3 promoter rather than the lncRNA promoter (or rather, the two promoters are the same/overlap). It is unclear and not mentioned how these two possibilities could be distinguished conclusively. An example of an intergenic lncRNA would be a lot more convincing. It would also be good to show more than one example (additional examples can be in the supplement).

As suggested by the reviewer we took an example of long-range interaction involving an intergenic lncRNA: LINC01488. This lncRNA has been shown to be associated to breast cancer risk through, and function in DNA repair pathways. Here we show that the level of methylation at an enhancer ~42 kb upstream of the lncRNA promoter is associated to expression of the lncRNA in two patient cohorts (Oslo2 and TCGA). The enhancer is also a binding site for major regulators of estrogen signaling, ESR1, FOXA1, and GATA3, and is in direct chromatin interaction with the TSS of the lncRNA. Additionally, LINC01488 is in long-range interaction with its neighboring gene, CCND1, and studies have shown that knockdown of this lncRNA resulted in downregulation of CCND1 expression in the MCF7 cell line (PMID: 28777932). We

have now included this example of long-range interaction with an intergenic lncRNA and an mRNA as part of Figure 5, and revised the results (pages 13 and 14, lines 382-398) and the discussion (page 17, lines 484-499). MIR29BCHG and GATA3-AS1, which was previously in the main figure, are now included in supplementary figures.

Discussion:

- The discussion seems disproportionately long, with some more detailed description of results likely better suited in the Results section of the manuscript.

We revised the discussion according to the reviewer comment and both shortened and made the discussion more focused.

Reviewer #2 (Remarks to the Author):

The manuscript entitled 'Subtype and cell type specific expression of lncRNAs in breast cancer' by Sunniva Bjørklund et al. is a research article which addresses the associations between lncRNA expression and breast cancer clinicopathological features in two large population-based cohorts (SCAN-B) and the TCGA-BRCA cohort. They discovered three distinct clusters of lncRNAs using co-expression analysis of lncRNAs with protein coding genes. Then they identified the subtype specific transcription factor (TF) occupancy at their promoters, and integrated lncRNA expression with DNA methylation data to identify long range regulatory regions for lncRNA which were validated using ChiA-Pet-Pol2 loops. This research is novel and interesting. However, the following concerns need to be addressed.

Major concerns:

1 The author discovered the difference of lncRNA expression in different subtypes of breast cancer cells, but the author did not further explore the potential significance and validation of this result (clinical significance, auxiliary pathological typing judgment, guidance for treatment, prediction of patient prognosis, etc.).

We thank the reviewer for these suggestions. We performed Cox proportional hazards regression analysis in the SCAN-B cohort, with validation analysis in the TCGA cohort. To avoid the confounding factor of ER status we performed this analysis in ER+ and ER- patients separately. We identify 305 lncRNAs that are associated with overall survival of ER+ patients in the SCAN-B cohort, 4 of which were validated in TCGA, including the below mentioned MAPT-AS1. We have included the results of the survival analysis in Supplementary Table T2 and discussed these results in (page 6, lines 161-172, and page 10, lines 300-302)

I think the authors should at least detect these lncRNAs expression levels in different kinds of breast cancer cell lines to confirm that these novel lncRNAs are indeed the subtype and cell type specific expression lncRNAs in breast cancer.

We agree with the reviewer. For the revised version of the manuscript, we have performed lncRNA gene expression analysis from breast cancer cell lines representing different molecular subtypes. Although these cell lines are derived from very few patients, we were able to confirm that the Cluster1 (ESR1) lncRNAs were expressed at higher levels in Luminal A and Luminal B cell lines and have included this analysis in Supplementary Figure S7, and results section page 9 lines 268-281.

2 The authors use the SCAN-B and TCGA-BRCA database to analysis the lncRNAs. And they found several lncRNAs specific expression in a certain subtype of breast cancer, such as FOXCUT, MAPT-AS1, LINC00992. Several reports have demonstrated these lncRNAs functions in breast cancer. For instance, it has been reported that MAPT-AS1 is overexpressed in breast cancer but not in triple negative breast cancer (TNBC), and that high expression of MAPT-AS1 is correlated with better patient survival (Biochem Cell Biol. 2019 Apr;97(2):158-164. doi:10.1139/bcb-2018-0039.). The authors should cite these published papers to confirm the finding of different expression of lncRNAs in different subtypes of breast cancer. It will make the authors' bioinformatics analysis results much more solid. We thank the reviewer for pointing out these references, and we have included and discussed these references in the new section on Survival analysis on page 6 lines 165-168. Indeed, in our analyses included here MAPT1-AS1 was significantly associated with overall survival of ER+ patients in both cohorts.

Other minor concerns should be addressed:

1 The data sample information has reached 1097 in TCGA-BRCA of UCSC Xena browser (update time: 2021, March 31th). I suggest the authors update to the newest TCGA-BRCA database to analysis the lncRNAs expression. Indeed the different databases operate with slightly different patient and sample numbers in TCGA studies (i.e the PanCancer study of TCGA- PMID: 29625055). We have used the GDC as our source for rna-seq raw files and noted that some patients have several samples sequenced/several files for the same sample. We have removed FFPE samples, and in the case of duplicated sequencing files we have kept the file from the latest plate id as recommended by the Broad institute (<https://broadinstitute.atlassian.net/wiki/spaces/GDAC/pages/844334036/FAQ#FAQ-replicateFilteringQ%3AWhatdoyoudowhenmultiplealiquotbarcodesexistforagivensample%2Fportion%2Fanalytecombination%3F>). We included a list of TCGA IDs used in the analysis as Supplementary Table T9.

2 In figure 1c-d, the authors search for the overlap lncRNAs in the SCAN-B database and TCGA-BRCA database, while we notice that the ER/Her2 positive and negative samples which the authors showed in the two panels are much less than the two databases sample number (SCAN-B n=3455; TCGA-BRCA n=1095). So the authors need state the data cleanout standard in the two databases. Besides, it will be better if the authors can analysis the lncRNAs expression status in the samples both have ER and Her2 information.

We thank the reviewer for this comment and apologize that the Figure 1c-d has not been well enough described; the numbers here refer to lncRNAs that are up-regulated in the same subtype in both cohorts. To address this issue, we have now clarified the figure text. We have also added a sentence in the materials and methods on the data cleanout and the number of patients in each clinical group, which we also included in Supplementary Table T9.

REVIEWERS' COMMENTS:

Reviewer #1 (Remarks to the Author):

I appreciate that the authors have added a number of additional analyses as suggested to strengthen the manuscript. My previous comments have been sufficiently addressed, and I am happy to endorse the publication of this manuscript in Communications Biology.

Reviewer #2 (Remarks to the Author):

In the revised manuscript, the authors have added new data to answer my questions. I believe this research will give us a better understanding of the associations between lncRNAs expression and breast cancer clinicopathological features. I recommend it publish on Communications Biology.